# 3D Indoor Instance Segmentation in an Open-World

**Mohamed El Amine Boudjoghra**[1]**, Salwa K. Al Khatib**[1]**, Jean Lahoud**[1]**,**
**Hisham Cholakkal**[1]**, Rao Muhammad Anwer**[1,2]**, Salman Khan**[1,3]**, Fahad Shahbaz Khan**[1,4]

[1]Mohamed Bin Zayed University of Artificial Intelligence (MBZUAI),
[2]Aalto University, [3]Australian National University, [4]Linköping University

```
{mohamed.boudjoghra, salwa.khatib, jean.lahoud,
hisham.cholakkal, rao.anwer, salman.khan, fahad.khan}@mbzuai.ac.ae
```

## Abstract

Existing 3D instance segmentation methods typically assume that all semantic classes to be segmented would be available during training and only seen categories are segmented at inference. We argue that such a closed-world assumption is restrictive and explore for the first time 3D indoor instance segmentation in an open-world setting, where the model is allowed to distinguish a set of known classes as well as identify an unknown object as unknown and then later incrementally learning the semantic category of the unknown when the corresponding category labels are available. To this end, we introduce an open-world 3D indoor instance segmentation method, where an auto-labeling scheme is employed to produce pseudo-labels during training and induce separation to separate known and unknown category labels. We further improve the pseudo-labels quality at inference by adjusting the unknown class probability based on the objectness score distribution. We also introduce carefully curated open-world splits leveraging realistic scenarios based on inherent object distribution, region-based indoor scene exploration and randomness aspect of open-world classes. Extensive experiments reveal the efficacy of the proposed contributions leading to promising open-world 3D instance segmentation performance. Code and splits are available at: https://github.com/aminebdj/3D-OWIS.

## 1 Introduction

3D semantic instance segmentation aims at identifying objects in a given 3D scene, represented by a point cloud or mesh, by providing object instance-level categorization and semantic labels. The ability to segment objects in the 3D domain has numerous vision applications, including robotics, augmented reality, and autonomous driving. Following the developments in the sensors that acquire depth information, a variety of datasets has been presented in the literature which provides instance-level annotations. In view of the availability of large-scale 3D datasets and the advances in deep learning methods, various 3D instance segmentation methods have been proposed in recent years.

The dependence of 3D instance segmentation methods on available datasets has a major drawback: a fixed set of object labels (vocabulary) is learned. However, object classes in the real world are plentiful, and many unseen/unknown classes can be present at inference. Current methods that learn on a fixed set not only discard the unknown classes but also supervise them to be labeled as background. This prevents intelligent recognition systems from identifying unknown or novel objects that are not part of the background. Given the importance of identifying unknown objects, recent works have explored open-world learning setting for 2D object detection [18, 11, 28, 33]. In the open-world setting, a model is expected to identify unknown objects, and once new classes are labeled, the new set is desired to be incrementally learned without retraining [18]. While previous methods have been mostly suggested for open-world 2D object detection, it is yet to be explored

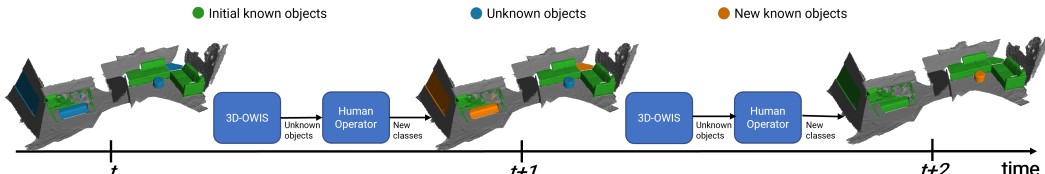

Figure 1: **3D instance segmentation in an open-world.** During each iterative learning phase, the model detects *unknown* objects, and a human operator gradually assigns labels to some of them and incorporates them into the pre-existing knowledge base for further training.

in the 3D domain. The main challenge lies in understanding how objects appear in 3D in order to separate them from the background and other object categories.

3D instance segmentation in the open world, illustrated in Fig. 1, offers more flexibility, allowing the model to identify unknown objects and request annotations for these novel classes from an oracle for further training. However, this approach presents several challenges: (i) the lack of annotations for unknown classes, necessitating quality pseudo-labeling techniques; (ii) the similarities between predicted features of known and unknown classes, requiring separation techniques for improved prediction; and (iii) the need for a more reliable objectness scoring method to differentiate between good and bad predicted masks for 3D point clouds.

In this work, we investigate a novel problem setting, namely open-World indoor 3D Instance Segmentation, which aims at segmenting objects of unknown classes while incrementally adding new classes. We define real-world protocols and splits to test the ability of 3D instance segmentation methods to identify unknown objects. In the proposed setup, unknown object labels are also added incrementally to the set of known classes, akin to real-world incremental learning scenarios. We propose an unknown object identifier with a probability correction scheme that enables improved recognition of objects. To the best of our knowledge, we are the first to explore 3D instance segmentation in an open-world setting. The key contributions of our work are:

- We propose the first open-world 3D indoor instance segmentation method with a dedicated mechanism for accurate identification of 3D unknown objects. We employ an auto-labeling scheme to generate pseudo-labels during training and induce separation in the query embedding space to delineate known and unknown class labels. At inference, we further improve the quality of pseudo-labels by adjusting the probability of unknown classes based on the distribution of the objectness scores.

- We introduce carefully curated open-world splits, having known vs. unknown and then incremental learning over the span of 200 classes, for a rigorous evaluation of open-world 3D indoor segmentation. Our proposed splits leverage different realistic scenarios such as inherent distribution (frequency-based) of object classes, various class types encountered during the exploration of indoor areas (region-based), and the randomness aspect of object classes in the open-world. Extensive experiments reveal the merits of the proposed contributions towards bridging the performance gap between our method and oracle.

## 2 Related Work

**3D semantic instance segmentation:** The segmentation of instances in 3D scenes has been approached from various angles. Grouping-based or clustering-based techniques use a bottom-up pipeline by learning an embedding in the latent space to help cluster the object points. [4, 13, 14, 17, 20, 21, 34, 38]. Proposal-based methods work in a top-down fashion, first detecting 3D bounding boxes, then segmenting the object region within the box [10, 15, 22, 36, 37]. Recently, spurred by related 2D work [5, 6], the transformer design [31] has also been applied for the purpose of segmenting 3D instances [29, 30]. Other methods present weakly-supervised alternatives to methods that use dense annotations in order to lower the cost of annotating 3D data [7, 16, 35]. While all these methods aim to improve the quality of 3D instance segmentation, they are trained on a known set of semantic labels. On the other hand, our proposed method aims at segmenting objects with both known and unknown class labels.

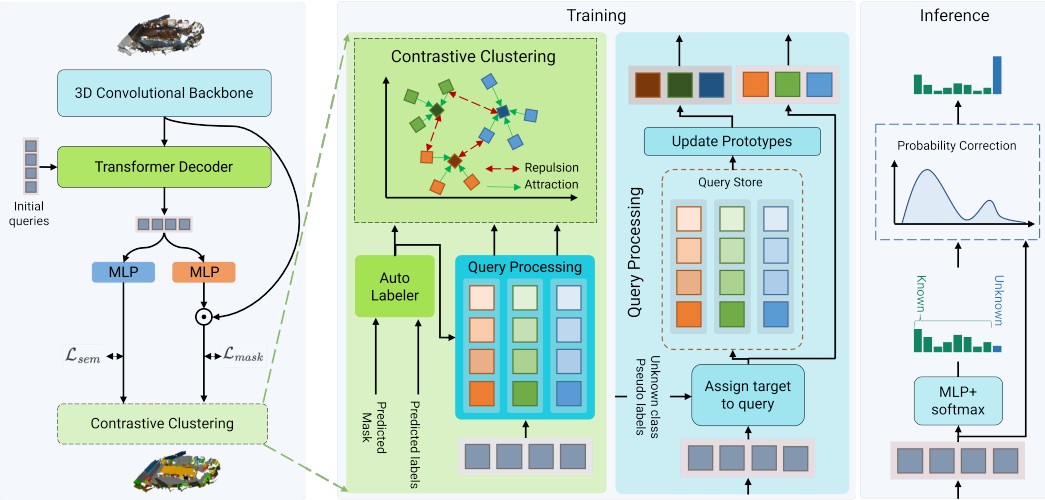

Figure 2: **Proposed open-world 3D instance segmentation pipeline.** From left to right: 3D instance segmentation model, where the point cloud goes through a 3D convolutional backbone. The extracted feature maps are used in the transformer decoder to refine some initial queries, which then pass through two MLPs to generate label and mask predictions. The Contrastive Clustering block takes the refined queries, the prediction masks, and labels to further process the queries by assigning a target or an *unknown* pseudo label in the Query Processing module, and then storing them in a Query Store to finally update the class prototypes, which are finally used for contrastive clustering. During inference, the queries are used to correct the probability of the predicted labels based on their reachability to the *known* class prototypes.

**Open-world object recognition:** Open-world object recognition was introduced in [2], where the Nearest Mean Classifier was extended for an open-world setting. In the direction of open-world object detection, many studies [41, 18, 11, 25] have been conducted in the past. In[18], pseudo-labels for the unknowns are generated to perform contrastive clustering during training for a better unknown-known classes separation, where an energy-based unknown class identifier was proposed to detect the unknown classes, based on the energy of the logits from the known classes. For incremental learning, they adopted exemplar replay to alleviate catastrophic forgetting of old classes. In the same task as [18], [11] used a transformer-based model and proposed another way of unknown pseudo-labels generation, by using a new method of objectness estimation, and introduced a foreground objectness branch that separates the background from the foreground. For the task of outdoor 3D point cloud semantic segmentation, [3] proposed a model that predicts old, novel, and unknown classes from three separate classification heads. The latter is trained on the labels of the known classes and pseudo-labels for old classes generated by the same model to alleviate catastrophic forgetting, while the unknown class is assigned the second-highest score for a better unknown class segmentation. Other methods proposed in [40, 12, 39], primarily focus on enhancing the generalizability of 3D models for novel classes by leveraging supervision from 2D Vision Language Models for object recognition and 3D semantic segmentation tasks. However, these approaches exhibit several limitations, including (i) The 3D model's performance becomes dependent on the 2D Vision Language model. (ii) The 3D geometric properties of unseen objects in the training data are neglected during the training process. (iii) There exists no avenue for enhancing the model's performance on novel classes in cases where new labels are introduced.(iv) The training process necessitates pairs of images and corresponding 3D scenes.

## 3   Closed-world 3D Instance Segmentation

We adopted the state-of-the-art 3D instance segmentation model Mask3D [29] as our baseline. The latter is a hybrid model that combines Convolutional Neural Networks (CNNs) with transformers to learn class-agnostic masks and labels for instance separation. The backbone of Mask3D is CNN-based and used to extract feature maps from multiple levels. Meanwhile, the decoder is transformer-based and used to refine $n_Q \in \mathbb{N}$ instance queries $Q = \{q_j \in \mathbb{R}^D \mid j \in (1, ..., n_Q)\}$, using the extracted

feature maps. The learning scheme consists of a Cross-entropy loss for learning semantic class labels and binary cross-entropy loss for learning instance masks during training.

# 4 Open-World 3D Instance Segmentation

## 4.1 Problem formulation

We start by formulating the problem setting of open-world 3D instance segmentation. At a Task $\mathcal{T}^t$, there exists a set of *known* object categories $\mathcal{K}^t = \{1, 2, .., C\}$ and a set of *unknown* object categories $\mathcal{U}^t = \{C + 1, ...\}$ that may exist on inference time. The training dataset $\mathcal{D}^t = \{\mathbf{X}^t, \mathbf{Y}^t\}$ includes samples from the classes $\mathcal{K}^t$. The input set $\mathbf{X}^t = \{\mathbf{P}_1, .., \mathbf{P}_M\}$ is made of $M$ point clouds, where $\mathbf{P}_i \in \mathbb{R}^{N \times 3}$ is a quantized point cloud of $N$ voxels each carrying average RGB color of the points within. The corresponding labels are $\mathbf{Y}^t = \{\mathbf{Y}_1, .., \mathbf{Y}_M\}$, where $\mathbf{Y}_i = \{\mathbf{y}_1, .., \mathbf{y}_k\}$ encodes $k$ object instances. Each object instance $\mathbf{y}_i = [\mathbf{B}_i, l_i]$ represents a binary mask $\mathbf{B}_i \in \{0, 1\}^N$ and a corresponding class label $l_i \in \mathcal{K}^t$.

In our problem setting, $\mathcal{M}_C$ is a 3D instance segmentation model that is trained on $C$ object categories, and, on test time, can recognize instances from these classes, in addition to instances from new classes not seen during training by classifying them as *unknown*. The detected *unknown* instances can be used by a human user to identify a set of $n$ new classes not previously trained on, which can be incrementally added to the learner that updates itself to produce $\mathcal{M}_{C+n}$ without explicitly retraining on previously seen classes. At this point in Task $\mathcal{T}^{t+1}$, the *known* class object categories are $\mathcal{K}^{t+1} = \mathcal{K}^t \cup \{C + 1, .., C + n\}$. This process repeats throughout the lifespan of the instance segmentation model, continuously improving itself by incorporating new information from new classes until it reaches its maximum capacity of classes it can learn. In the rest of the paper, We assign the *unknown* class a label $\mathbf{0}$.

## 4.2 Open-world scenarios

In order to simulate different realistic scenarios that might be encountered in an open-world, we propose three different ways of grouping classes under three tasks. These scenarios split scenes based on the inherent distribution (frequency-based) of object classes, the various classes encountered during the exploration of various indoor areas (region-based), and the randomness aspect of object classes in the open world.

Table 1: **The statistics of each split across the three tasks.** The number of known classes per task is reported along with the count of instances (3D objects) in the training and validation set, we also show the number of non-empty scenes used during training and validation.

|  | Split A | | | Split B | | | Split C | | |
|---|---|---|---|---|---|---|---|---|---|
|  | Task 1 | Task 2 | Task 3 | Task 1 | Task 2 | Task 3 | Task 1 | Task 2 | Task 3 |
| Classes count | 64 | 68 | 66 | 73 | 55 | 70 | 66 | 66 | 66 |
| Train instances | 24224 | 3791 | 1612 | 15327 | 8177 | 6123 | 13483 | 8239 | 7905 |
| Validation instances | 6539 | 1000 | 428 | 4177 | 2261 | 1529 | 3776 | 2102 | 2089 |
| Train scenes | 1201 | 924 | 627 | 1201 | 1002 | 895 | 1169 | 1089 | 1159 |
| Validation scenes | 312 | 242 | 165 | 312 | 264 | 236 | 307 | 273 | 300 |

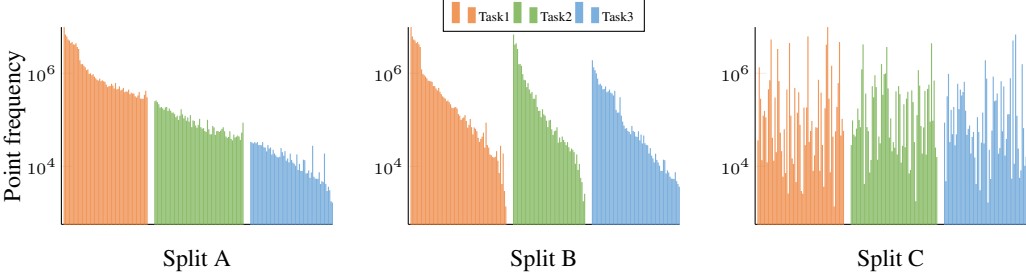

Figure 3: Point-wise count for each class across the three tasks under the three open-world scenarios

**Split A (Instance frequency-based):** We introduce a split that leverages the inherent distribution of objects, with *known* classes being more prevalent than *unknown* categories. Task $\mathcal{T}^1$ encompasses all the head classes as defined in the ScanNet200 benchmark [8, 27], while tasks $\mathcal{T}^2$ and $\mathcal{T}^3$ group the common and tail classes, respectively. This division allows us to effectively capture the varying frequency and significance of object categories within the dataset.

**Split B (Region-based):** In this split, our objective is to replicate the diverse class types encountered during indoor exploration.We argue that a perfect model for a robot moving indoors should segment both classes it knows and classes it hasn't seen before. Additionally, it should keep learning and getting better at segmenting new classes over time. This partition draws inspiration from the sequence of classes that a robot might encounter when navigating indoors. To achieve this, we group classes that are likely to be encountered initially when accessing an indoor space and share similarities in scenes. Initially, we assign each class to a specific scene where it predominantly occurs. Subsequently, we divide the classes into three distinct groups, corresponding to the three tasks.

**Split C (Random sampling of classes):** This third split introduces a different challenge inspired by the randomness aspect of the open-world, where tasks can exhibit random levels of class imbalance. To create this split, we randomly shuffled the classes and sampled without replacement, selecting 66 classes three times for each task.

### 4.3 Generating pseudo-labels for the unknown classes

Because of the wide range of classes in an open-world setting, the auto-labeler is used as an alternative to manual labeling. The former makes use of the existing target labels from the available ground truth classes (*known* classes) to generate pseudo-labels for the *unknown* class in the process of training. In [18], the model is assumed to be class agnostic, where *unknown* objects are predicted as *known* with high confidence. As a result, the authors of the paper proposed to use the predictions with top-k confidence scores that do not intersect with the ground truth as pseudo-labels for the *unknown* class. In our study, we show that top-k pseudo-label selection can severely harm the performance of the model on the *known* and *unknown* classes. Hence, we propose a Confidence Thresholding (**CT**) based selection of pseudo-labels. We show that the performance on the *known* and the *unknown* classes increases by a large margin in terms of mean Average Precision (mAP).

The *auto-labeler* unit, depicted in Fig. 2, is used for *unknown* pseudo-labels generation. It takes a set of predicted binary masks $\mathbf{B} = \{\mathbf{B}_i \mid i \in (1, ..., n_Q)\}$, where $n_Q$ is the number of queries, $\mathbf{B}_i = \mathbb{1}(M_i > 0.5)$ is a mask from a single query, and $M_i = \{m_{i,j} \in [0, 1] \mid j \in (1, ..., N)\}$ is a heat map measuring the similarity between a query $q_j \in \mathbb{R}^D$ and the features of $N$ voxels extracted from the high-resolution level in the backbone.

Moreover, each query $q_j$ encodes semantic information and can generate a class prediction $\mathbb{P}_{cls}(q_j) = \{\mathbb{P}_{cls}(c; q_j) \mid c \in (0, 1, ..., |\mathcal{K}^t|)\}$ using a classification head (refer to Fig. 2). Subsequently, the objectness confidence score is assigned to predictions following Eq 1.

$$s_j = s_{cls,j} \cdot \frac{M_j \cdot \mathbb{1}(M_j > 0.5)^T}{|\mathbb{1}(M_j > 0.5)|_1} \tag{1}$$

where $s_{cls,j} \in \mathbb{R}$ is the max output probability from the classification head $\mathbb{P}_{cls}(q_j)$, and $\mathbb{1}$ is the indicator function. After scoring the predictions, the auto-labeler returns $m$ pseudo-labels $\tilde{\mathbf{Y}} = \{\tilde{\mathbf{y}}_i = [\tilde{\mathbf{B}}_i, \mathbf{0}] \mid i \in (1, ..., m)\}$ with confidence above a threshold and has a low IoU with the *known* classes' target masks.

### 4.4 Query target assignment and contrastive clustering

Similar to [18], we utilize contrastive clustering to enhance the separation of classes within the query embedding space. To achieve this, we employ a set of query prototypes denoted as $\mathcal{Q}_p = \{\mathbf{q}_i \in \mathbb{R}^D \mid i \in (0, 1, .., |\mathcal{K}^t|)\}$, where $\mathbf{q}_0$ denotes the prototype of the class *unknown*. We apply a contrastive loss that encourages queries with similar classes to be attracted to their respective prototypes while pushing them away from those representing negative classes, as illustrated in Fig. 2. Since the queries are used to determine the class of the objects (see Fig. 2 inference block), the class prototypes are expected to hold general semantic knowledge of their corresponding classes.

*Hungarian matching* is performed in the *Assign target to query* module, depicted in Fig. 2, where the indices of prediction-target are used to assign a label to the queries used to generate the matched prediction. The labeled queries are then stored in a *query store* $\mathcal{Q}_{store}$, which represents a queue with a maximum capacity. This queue is employed to update the query prototypes $\mathcal{Q}_p$ using an exponential moving average.

Hinge embedding loss is utilized according to Eq 2. This loss ensures that queries belonging to the same class denoted as $q_c$, are pulled towards their corresponding class prototype $\mathbf{q}_c$, while being pushed away from other prototypes representing different classes.

$$\mathcal{L}_{cont}(q_c) = \sum_{i=0}^{|\mathcal{K}^t|} \ell(q_c, \mathbf{q}_i) \tag{2}$$

$$\ell(q_c, \mathbf{q}_i) = \begin{cases} ||q_c - \mathbf{q}_i||_2 & i = c \\ \max(0, \Delta - ||q_c - \mathbf{q}_i||_2) & i \neq c \end{cases}$$

where $\Delta$ is the margin of the contrastive clustering.

## 4.5 Reachability-based probability correction (PC)

In [23], an architecture that can deal with long-tail distribution and *unknown* class prediction for open-world object recognition was proposed, where *unknown* classes are assumed to be very different in color and texture from the *known* classes without prior on the *unknown* classes. However, we show in Fig. 6 that many *unknown* instances hold similar features to the *known* ones.

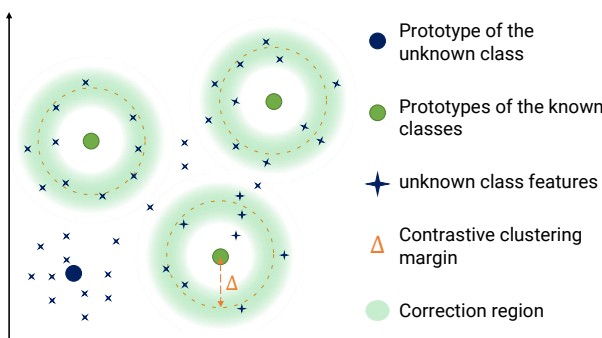

In our method, we relax the strict assumption of high dissimilarity of *unknown* and *known* classes and correct the predicted output probability following two characteristics of a feature

Figure 4: Illustration of the region in the query embedding space where the class probability is corrected.

from an *unknown* object: (1) it has to be far from the nearest *known* class, as features of the class *unknown* are expected to be pushed away from the prototypes of the *known* classes, after applying constructive clustering, and (2) the feature should correspond to an object that is not a *known* class. We show that applying this approach during inference boosts the performance of the model on the *unknown* class considerably by compensating for the weak pseudo-labels provided by the auto-labeler.

Our probability correction scheme is the following

$$\mathbb{P}(\mathbf{0}; q_j) = \mathbb{P}_{cls}(\mathbf{0}; q_j) \cup \mathbb{P}_{corr}(\mathbf{0}; q_j) \tag{3}$$

where $\mathbb{P}_{cls}$ is the probability from the classification head, and $\mathbb{P}_{corr}$ is the correction probability. We base our intuition on the fact that *unknown* classes have high objectness scores, which makes them not too far from the prototypes of the *known* classes. To model this behavior we choose

$$\mathbb{P}_{corr}(\mathbf{0}; q_j) = \mathbb{P}_{corr}(\mathbf{0}; o, q_j) \cdot \mathbb{P}_{corr}(o; q_j)$$

where $\mathbb{P}_{corr}(o; q_j)$ is the likelihood of the query to correspond to an object that is not *known* (either background or true *unknown*). Since the query prototypes encode class-specific information we propose the following method to measure the objectness of a query given all prototypes from the *known* classes, where it assigns a high objectness probability if it is close to only a few *known* classes. This probability distribution defines the objectness of *unknown* objects around a certain boundary from the prototypes as follows.

$$\mathbb{P}_{corr}(o; q_j) = 1 - \sum_{k=1}^{|\mathcal{K}^t|} \mathbb{P}_{cls}(k; q_j)$$

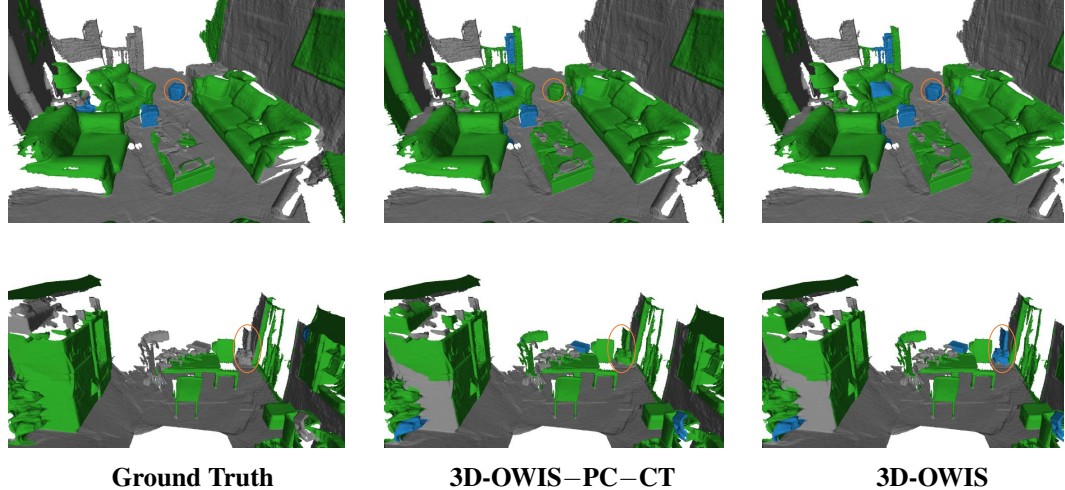

| **Ground Truth** | **3D-OWIS−PC−CT** | **3D-OWIS** |

Figure 5: **Qualitative results for 3D instance segmentation results on some ScanNet200 validation scenes**. Points highlighted in blue belong to *unknown* classes and those highlighted in green belong to *known* classes. We show the performance of our model in retrieving the *unknown* class objects compared to **3D-OWIS−PC−CT** for the three scenes.

while $\mathbb{P}_{corr}(\mathbf{0}; o, q_j)$ is the probability of the query being an *unknown* object, which has a high value the further it is from the nearest prototype of the *known* classes.

$$\mathbb{P}_{corr}(\mathbf{0}; o, q_j) = \sigma\left(\frac{\gamma(q_j) - a}{b}\right); \qquad \gamma(q_j) = \min_{\mathbf{q}_i} ||q_j - \mathbf{q}_i||_2$$

Here $\sigma$ is the sigmoid function, $\gamma(q_j)$ is the reachability of the query $q_j$, $\mathbf{q}_i$ is the prototype of the $i^{th}$ class, and $a, b$ are the shift and scale of the sigmoid function that assure $\mathbb{P}_{corr}(\mathbf{0}; o, q_j, \gamma(q_j) = 0) = 0.05$ and $\mathbb{P}_{corr}(\mathbf{0}; o, q_j, \gamma(q_j) = \frac{\Delta}{2}) = 0.95$, for a contrastive clustering margin $\Delta$.

We finally normalize the probabilities from the classification head of the *known* classes as follows

$$\mathbb{P}(c; q_j) = \frac{\mathbb{P}_{cls}(c; q_j)}{\sum_{l \in \mathcal{K}^t} \mathbb{P}_{cls}(l; q_j)}(1 - \mathbb{P}(\mathbf{0}; q_j))$$

### 4.6 Alleviating catastrophic forgetting for incremental learning

Following the success of exemplar replay in avoiding catastrophic forgetting of the old classes during incremental learning for object detection [18, 11, 41], we adopt it for the task of incremental learning in 3D instance segmentation where we use exemplars from the classes of the previous task to fine-tune the model trained on the novel classes. In our setting, we use the same dataset for the three tasks and mask the classes of the previous task when training on the novel classes from the current task. As a result, the novel classes of the current task might be encountered again when replaying the exemplars from the previous task, as the same scenes are being used in fine-tuning.

## 5 Experiments

### 5.1 Open-world evaluation protocol

We use our proposed splits of classes which mimic the challenges that are mostly faced in the open-world to ensure a strict performance evaluation for 3D instance segmentation models.

**Evaluation metrics.** We adopt three common evaluation metrics, *wilderness impact* (WI) [9], *absolute open set error* (A-OSE) [26], and the *recall of the unknown classes* (U-Recall) [1, 24, 11]

Table 2: **State-of-the-Art comparison for 3D-OWIS model.** We show a comparison of performance under the three open-world scenarios, where **3D-OWIS−PC − CT** is our model **3D-OWIS** without Probability Correction (**PC**) and Confidence Thresholding (**CT**). We rely on the metrics used in the open-world literature, A-OSE which quantifies the number of unknown objects misclassified as one of the known classes, WI which measures the impact of the unknown class on the precision of the model on the known classes, and the U-Recall to evaluate the model's ability to recover the unknown objects. We show that **3D-OWIS** performs remarkably better than the other models under all scenarios when dealing with the known classes, and superior performance in split A and B, and slightly less performance in split C when handling the unknown objects. We also provide a closed-setting comparison between Mask3D and Oracle (**Ours** with access to unknown labels).

| Task IDs (→) | Task 1 | | | | | Task 2 | | | | | | Task 3 | | |
|---|---|---|---|---|---|---|---|---|---|---|---|---|---|---|
| | WI | A-OSE | U-Recall | mAP (↑) | | WI | A-OSE | U-Recall | mAP (↑) | | | mAP (↑) | | |
| | (↓) | (↓) | (↑) | Current known | All | (↓) | (↓) | (↑) | Previously known | Current known | All | Previously known | Current known | All |
| **Split A** | | | | | | | | | | | | | | |
| Oracle | 0.129 | 227 | 55.94 | 38.75 | 38.60 | 0.03 | 112 | 45.40 | 38.25 | 20.91 | 29.40 | 29.58 | 17.78 | 26.10 |
| Mask3D [29] | - | - | - | 39.12 | 39.12 | - | - | - | 38.30 | 20.57 | 29.15 | 28.61 | 18.33 | 25.58 |
| 3D-OW-DETR [11] | 0.547 | 721 | 22.14 | 35.56 | 35.05 | 0.282 | 253 | 26.24 | 18.18 | 13.62 | 15.76 | **21.56** | 08.38 | 17.67 |
| 3D-OWIS−PC − CT | 1.589 | 707 | 30.72 | 37.50 | 37.00 | **0.000** | **4** | 04.75 | 11.00 | **17.30** | 14.10 | 21.40 | 08.00 | 17.50 |
| **Ours: 3D-OWIS** | **0.397** | **607** | **34.75** | **40.2** | **39.7** | 0.007 | 126 | **27.03** | **29.40** | 16.40 | **22.70** | 20.20 | **15.20** | **18.70** |
| **Split B** | | | | | | | | | | | | | | |
| Oracle | 1.126 | 939 | 70.31 | 24.57 | 24.80 | 0.180 | 441 | 73.16 | 25.50 | 20.30 | 23.40 | 23.40 | 30.40 | 26.00 |
| Mask3D [29] | - | - | - | 23.48 | 23.48 | - | - | - | 21.81 | 18.91 | 20.37 | 24.20 | 29.22 | 26.06 |
| 3D-OW-DETR [11] | 3.229 | 1935 | 17.18 | 20.00 | 19.73 | 2.053 | 1389 | **33.31** | 12.36 | 13.86 | 12.93 | 07.27 | 18.96 | 11.62 |
| 3D-OWIS−PC − CT | **3.133** | 1895 | 21.67 | 18.94 | 18.70 | 3.169 | 1081 | 26.63 | 18.00 | 16.40 | 17.20 | 17.30 | 20.10 | 18.30 |
| **Ours: 3D-OWIS** | 3.684 | **1780** | **24.79** | **23.60** | **23.30** | **0.755** | **581** | 24.21 | **18.70** | **17.30** | **17.90** | **18.70** | **24.60** | **20.90** |
| **Split C** | | | | | | | | | | | | | | |
| Oracle | 1.039 | 651 | 71.61 | 23.30 | 23.6 | 0.249 | 591 | 62.83 | 20.50 | 18.40 | 19.60 | 25.30 | 28.20 | 26.30 |
| Mask3D [29] | - | - | - | 20.82 | 21.15 | - | - | - | 22.67 | 26.67 | 24.13 | 25.41 | 25.21 | 25.35 |
| 3D-OW-DETR [11] | 1.463 | 1517 | 13.00 | 14.81 | 14.59 | 1.330 | 847 | **16.04** | 08.00 | 17.41 | 12.40 | 08.81 | 15.63 | 11.01 |
| 3D-OWIS−PC − CT | 2.901 | 1752 | **15.66** | 15.00 | 14.80 | 1.799 | 666 | 15.99 | 13.50 | 19.70 | 16.40 | 17.50 | **17.70** | 17.50 |
| **Ours: 3D-OWIS** | **0.419** | **1294** | 14.34 | **18.00** | **17.60** | **0.152** | **303** | 15.80 | **13.90** | **22.20** | **17.80** | **17.80** | 17.70 | **17.80** |

to evaluate the performance of our model on the *unknown* classes and to provide a fair comparison with and without contributions. For the *known* classes, we use mean Average Precision (mAP). WI measures the impact of the *unknown* classes on the precision of the model at a specific confidence level. Ideally, WI is nil, i.e., there are no *unknown* objects predicted as *known*. For our evaluation, we report WI at 0.5 confidence. It can be computed as follows: $\text{WI} = \frac{P_{\mathcal{K}}}{P_{\mathcal{K} \cup \mathcal{U}}} - 1$.

We also report A-OSE, which represents the count of *unknown* instances misclassified as one of the *known* classes, and the U-Recall at 0.5 IoU, which reflects the ability of the model to recover *unknown* objects.

## 5.2   Implementation details

We adapt Mask3D [29] for the task of open-world instance segmentation. We add an extra prediction output for the *unknown* class. In training, we assign an *ignore* label to the classes of the future and previous tasks, while we keep the labels of the previous task and assign an *unknown* class label to the classes of the future task during evaluation. For contrastive clustering, we use the indices obtained after matching the predictions with the target using *Hungarian matching* to assign a label to the queries and store them in the *Query Store* $\mathcal{Q}_{store}$. The store is then averaged per class and used to periodically update the prototypes every 10 iterations for the hinge loss computation. Finally, we use 40 exemplars per class on average for incremental learning. The classes from the current task are kept during class exemplar replay since we are using the same dataset for the three tasks.

## 5.3   Open-world results

Table 2 provides a comprehensive performance comparison between the Oracle, our implementation of [11] as 3D-OW-DETR, **3D-OWIS**, and **3D-OWIS−PC − CT** when excluding the Probability Correction (**PC**) and Confidence Thresholding (**CT**) components. Across all scenarios and tasks,

Table 3: **Extensive ablation of the added components.** We perform the ablation by adding Probability Correction (**PC**) and Confidence Thresholding (**CT**) components to **3D-OWIS**−**PC** − **CT**. We conduct the performance comparison in terms of mAP , U-Recall , WI , and A-OSE . Even though **3D-OWIS** is performing well in retrieving the *unknown* classes without **PC** and **CT**, which is reflected by the high U-Recall , it is still performing poorly on the *known* classes, based on the high WI and A-OSE . This negative impact on the *known* classes accumulates over the tasks and results in further reduction in mAP. When adding the **CT**, the performance on the *known* classes improves considerably and remains consistent throughout the incremental learning process. Probability correction (**PC**) significantly improves the U-Recall in all cases. Even though the latter shows lower performance in terms of WI and A-OSE , the overall mAP slightly improves or remains higher with a large margin compared **3D-OWIS**−**PC** − **CT**. This shows that adding **PC** and **CT** gives the best compromise in performance on both *known* and *unknown* classes.

| Task IDs (→) | | | Task 1 | | | | | Task 2 | | | | | | Task 3 | | |
|---|---|---|---|---|---|---|---|---|---|---|---|---|---|---|---|---|
| | | | WI | A-OSE | U-Recall | mAP (↑) | | WI | A-OSE | U-Recall | mAP (↑) | | | mAP (↑) | | |
| w/ Finetuning | CT | PC | (↓) | (↓) | (↑) | Current known | All | (↓) | (↓) | (↑) | Previously known | Current known | All | Previously known | Current known | All |
| **Split A** | | | | | | | | | | | | | | | | |
| × | × | × | 1.589 | 707 | 30.72 | 37.50 | 37.00 | 0.870 | 321 | 19.42 | 00.00 | 16.74 | 08.40 | 00.00 | 09.30 | 02.80 |
| × | ✓ | × | 0.237 | 443 | 30.00 | 40.30 | **39.70** | 0.306 | 129 | 14.96 | 00.00 | **21.00** | 10.50 | 00.00 | **17.45** | 05.20 |
| ✓ | × | × | 1.589 | 707 | 30.72 | 37.50 | 37.00 | **0.000** | **4** | 04.75 | 11.00 | 17.30 | 14.10 | **21.40** | 08.00 | 17.50 |
| ✓ | ✓ | × | **0.237** | **443** | 30.00 | **40.30** | 39.70 | 0.004 | 102 | 23.62 | 29.22 | 15.80 | 22.30 | 19.70 | 15.70 | **18.50** |
| ✓ | ✓ | ✓ | 0.398 | 607 | **34.75** | 40.2 | **39.70** | 0.007 | 126 | **27.03** | 29.40 | 16.40 | **22.70** | No unknown labels for evaluation | | |
| **Split B** | | | | | | | | | | | | | | | | |
| × | × | × | 3.133 | 1895 | 21.67 | 18.94 | 18.70 | 1.82 | 829 | 17.20 | 00.00 | 15.40 | 06.60 | 00.00 | 20.20 | 07.50 |
| × | ✓ | × | 2.147 | 21.70 | 21.70 | 23.80 | 23.50 | 1.563 | **375** | 13.08 | 00.00 | **18.30** | 07.90 | 00.00 | **25.40** | 09.40 |
| ✓ | × | × | 3.219 | 1905 | 21.70 | 18.94 | 18.70 | 3.169 | 1081 | 26.63 | 18.00 | 16.40 | 17.20 | 17.30 | 20.10 | 18.30 |
| ✓ | ✓ | × | **2.147** | **1397** | 21.70 | **23.80** | **23.50** | 0.466 | 413 | 20.90 | 18.60 | 16.90 | 17.70 | **18.50** | 24.20 | **20.60** |
| ✓ | ✓ | ✓ | 3.684 | 1780 | **24.79** | 23.6 | 23.30 | 0.755 | 581 | **24.21** | **18.70** | 17.30 | **17.90** | No unknown labels for evaluation | | |
| **Split C** | | | | | | | | | | | | | | | | |
| × | × | × | 2.901 | 1752 | 15.66 | 15.00 | 14.80 | 6.294 | 857 | 11.05 | 0.00 | 15.70 | 07.50 | 00.00 | 14.60 | 04.70 |
| × | ✓ | × | 0.227 | 828 | 11.44 | 18.70 | 18.40 | 1.361 | 365 | 10.16 | 00.00 | 19.50 | 09.40 | 00.00 | **19.10** | 6.20 |
| ✓ | × | × | 2.901 | 1752 | **15.66** | 15.00 | 14.80 | 1.799 | 666 | **15.99** | 13.50 | 19.70 | 16.40 | 17.50 | 17.70 | 17.50 |
| ✓ | ✓ | × | **0.227** | **828** | 11.44 | **18.70** | 18.40 | **0.088** | **208** | 12.63 | 14.50 | 22.10 | 18.00 | **17.80** | 17.70 | **17.80** |
| ✓ | ✓ | ✓ | 0.419 | 1294 | 14.34 | 18 | 17.60 | 0.152 | 303 | 15.80 | 13.90 | **22.20** | 17.80 | No unknown labels for evaluation | | |

**3D-OWIS**−**PC** − **CT** consistently exhibits inferior performance in terms of mAP. Additionally, it demonstrates considerably lower U-Recall performance in splits A and B, with slightly higher

performance in split C. Of particular note, our **3D-OWIS** demonstrates remarkable proficiency in preserving knowledge of the previous classes after fine-tuning. This proficiency is attributed to better pseudo-label selection for the *unknown* classes. **3D-OWIS** outperforms **3D-OWIS**−**PC** − **CT** in most cases while minimizing the impact of the *unknown* classes on the *known* classes, as evidenced by lower WI and A-OSE scores and higher mAP.

Table 4 presents a comparison between our model, **3D-OWIS**, and our implementation of two methods, GGN [32] and OLN [19]. For both models, we adapt Mask3D and train it with mask loss only for OLN. In the case of GGN, we train a Minkowski backbone to predict affinity maps and use Connected Components to generate class-agnostic proposals. These results underscore the effectiveness and potential of our approach in addressing the three proposed open-world challenges.

## 5.4 Incremental learning results

Our model's performance in incremental learning is evaluated based on its ability to preserve knowledge from previous classes. With the utilization of exemplar replay, the **3D-OWIS** model demonstrates significant improvement on previous classes mAP. Table 2 presents the results, indicating that our model consistently outperforms the others in terms of mean Average Precision (mAP) for the previous classes in all cases.

## 5.5 Discussion and analysis

**Ablation study.** We show in Table 3 that **3D-OWIS−PC − CT** model performs poorly on the *known* classes because of the high number of low-quality pseudo-labels generated by *Auto-labeler*, which is also explained by the high value of *Wilderness Impact* and *Absolute open set error*. The U-Recall drops considerably when fine-tuning the **3D-OWIS−PC − CT**, while the WI and A-OSE either decrease or increase with the mAP on the *unknown*. On the other hand, our model limits the training only to the best pseudo-labels, which maintain good performance on the *known* classes in all cases, before and after fine-tuning, and also achieve results on the *unknown* class comparable to the **3D-OWIS−PC − CT** in most of the cases. Adding the probability correction module helps in improving the U-Recall while keeping the mAP of the *known* classes much above the **3D-OWIS−PC − CT**. However, it results in an increase in WI and A-OSE because of the increase of false positives in the *known* classes.

Table 4: **Open-world instance segmentation comparison**. We provide the results of our implementation of two methods for 2D open-world instance segmentation models. We show that our model performs comparatively better than others across all metrics.

| Split A | | | | | |
|---|---|---|---|---|---|
| **Task ID** | | **Task 1** | | | |
| | WI | A-OSE | U-Recall | mAP (↑) | |
| | (↓) | (↓) | (↑) | Current known | All |
| 3D-GGN [32] | 15.68 | 1452 | 21.33 | 20.51 | 20.12 |
| 3D-OLN [19] | - | - | 02.45 | - | - |
| **Ours: 3D-OWIS** | **0.397** | **607** | **34.75** | **40.2** | **39.7** |

**tSNE analysis** The tSNE plot shown in Fig. 6 illustrates the below-par performance of the **3D-OWIS−PC − CT** in clustering the *unknown* classes, where most queries are still maintaining features representative of the *known* classes. This behavior is a result of the weak supervision of the *unknown* class, which shows the need for correcting the predictions, and explains the improvement in U-Recall when applying the probability correction with nil deterioration in the *known* classes mAP in most cases.

**Qualitative analysis.** Fig. 5 shows that 3D-OWIS is able to correctly identify background and *unknown* objects as *unknown*. Also note the second scene, where predictions are corrected from *known* to *unknown* without affecting the predictions of the *known* classes.

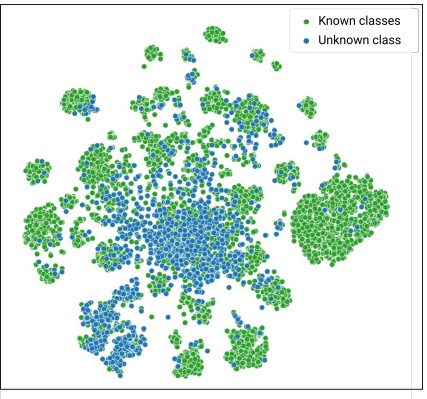

Figure 6: **tSNE visualization** of the queries for *known* & *unknown* classes

## 6 Limitations

Confidence Thresholding (**CT**) enhances the performance of the model on *known* classes; nonetheless, it diminishes the model's capacity to segment *unknown* classes, mainly due to its reliance on a smaller number of pseudo-labels during training. Additionally, the effectiveness of Probability Correction (**PC**) is contingent upon the inherent characteristics of the clusters within the *known* classes. In scenarios characterized by data imbalance, the performance of probability correction may deteriorate when applied to the undersampled classes.

## 7 Conclusion

In this paper, we address the challenge of 3D instance segmentation in open-world scenarios, which is a novel problem formulation. We propose an innovative approach that incorporates an *unknown* object identifier to detect objects not present in the training set. To facilitate evaluation and experimentation, we present three dataset splits of ScanNet200 based on different criteria for selecting *unknown* objects. Our experimental results demonstrate that our proposed *unknown* object identifier significantly improves the detection of *unknown* objects across various tasks and dataset splits. This work contributes to advancing the localization and segmentation of 3D objects in real-world environments and paves the way for more robust and adaptable vision systems.

**Acknowledgement** The computational resources were provided by the National Academic Infrastructure for Supercomputing in Sweden (NAISS), partially funded by the Swedish Research Council through grant agreement No. 2022-06725, and by the Berzelius resource, provided by Knut and Alice Wallenberg Foundation at the National Supercomputer Center.

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

# Appendix

## A    Scalability of 3D-OWIS

We show in Table 5 that **3D-OWIS** can accommodate a large number of classes without a major size increase

Table 5: Demonstrating the Scalability of **3D-OWIS** with Respect to the maximum number of classes it can learn.

| # of classes | 200 | 1000 | 5000 | 10000 | 50000 | 100000 |
|---|---|---|---|---|---|---|
| Size of **3D-OWIS** | 39.7M | 39.8M | 40.7M | 41.9M | 50.9M | 62.2M |

## B    Additional details on Split B

We utilize the 20 scene types present in the ScanNet200 dataset to distribute the 200 classes over the three tasks. Initially, we establish a notion of similarity between two scene types by assessing the extent of their shared classes. This similarity is quantified through the intersection over the union ($IoU$) metric, which measures the ratio of common classes to the total count of unique classes across both scenes. By employing this metric, we identify scene types that exhibit a substantial $IoU$, indicating a higher degree of similarity. The similarity matrix, depicted in Fig. 7, showcases the relationships between the 20 scene types within the ScanNet200 dataset.

Subsequently, we employed three criteria to group the classes: ($i$) the likelihood of encountering them first when accessing an indoor area, ($ii$) their affiliation with similar scene types, and ($iii$) the proximity in the number of known classes across tasks. By taking these factors into consideration, we arrived at the split of scenes presented in Table 6.

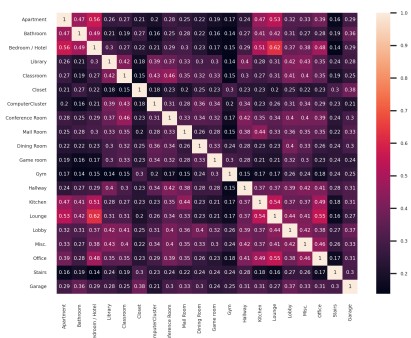

Figure 7: **Similarity matrix between the 20 scene types in ScanNet200 dataset.** We show the ratio of common classes to the total count of unique classes between two scene types.

Table 6: **Frequently occurring scene when training during the three tasks in Split B.** Scene types are grouped into tasks based on three criteria: ($i$) the likelihood of encountering the classes within the scene types when entering an indoor area, ($ii$) similarity of scene types containing the classes, and ($iii$) consistency in the overall number of classes within the scene types across all tasks. This grouping ensures a cohesive organization of scene types for effective evaluation of 3D instance segmentation models integrated with tasks such as robot navigation within indoor environments.

| Split B | | | | | |
|---|---|---|---|---|---|
| Task 1 | Task 2 | Task 3 | | | |
| Bedroom / Hotel | Kitchen | ComputerCluster | Mail Room | Game room | Office |
| Dining Room | Bathroom | Misc. | Hallway | Apartment | |
| Lounge | Closet | Gym | Classroom | Lobby | |
| | Garage | Library | Conference Room | Stairs | |

## C    Additional details on the experimentation

**Training:** We train the model on the entire ScanNet200 dataset for all tasks. In Task 1, objects belonging to the classes from Task 2 and Task 3 are masked, excluding them from the learning process. Moving to Task 2, we utilize the last saved checkpoint of the model from Task 1 as a starting point and mask the objects with labels that correspond to the current known classes of Task 1 and Task 3. This allows the model to focus solely on learning and distinguishing the specific objects associated with the current task. Finally, Task 3 builds upon the progress made in Task 2. We load the

latest checkpoint of the model from Task 2 and incorporate an exemplar replay. Similar to Task 2, the objects with labels belonging to the known classes in Task 1 and Task 2 are masked during training. This step further refines the model's understanding and discrimination abilities for the specific objects relevant to the current task.

**Evaluation:** To conduct the evaluation during a task, we assign the "*unknown*" label to the known classes from all the future tasks.

## D    Additional qualitative results

### D.1    Unknown objects identification

The qualitative results depicted in Fig. 10, 12, 13, and 11 highlight the superior performance of our contribution in retrieving unknown objects. Across the majority of scenes, our model consistently corrects the mispredicted unknown classes while preserving the accuracy of known objects, thus demonstrating its robustness and effectiveness.

### D.2    Learning novel classes

Fig. 8 and Fig. 9 illustrate the sequential process of learning novel classes after identifying unknown objects from the previous task. In Fig. 8, we demonstrate the effectiveness of our method in successfully retrieving unknown classes in all tasks. Additionally, in Fig. 9, we highlight the potential of exemplar replay in retaining knowledge of the old classes after learning the novel classes in Task 2 and Task 3.

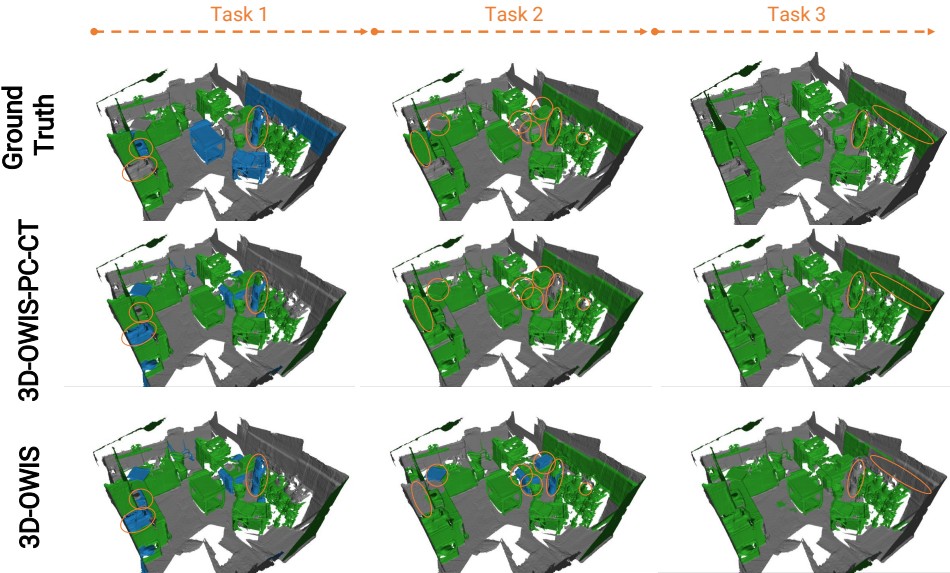

Figure 8: **Illustration of the process of unknown identification and learning novel classes.** We use orange circles to highlight the differences between **3D-OWIS** and **3D-OWIS−PC−CT**. The objects depicted in green represent the known classes, while those in blue represent the unknown objects. The gray objects correspond to the background. The qualitative results demonstrate that **3D-OWIS** outperforms **3D-OWIS−PC−CT** in retrieving unknown objects. Notably, **3D-OWIS** correctly identifies the background objects as unknown, whereas **3D-OWIS−PC−CT** misclassifies them as known objects.

Table 7: **Proposed distribution of ScanNet200 classes across tasks for each split.** We show the classes that are known when training the model during a specific task for the three splits.

| Split A | | | Split B | | | Split C | | |
| --- | --- | --- | --- | --- | --- | --- | --- | --- |
| Task 1 | Task 2 | Task 3 | Task 1 | Task 2 | Task 3 | Task 1 | Task 2 | Task 3 |
| tv stand | cushion | paper | alarm clock | guitar | bar | basket | ironing board | mattress |
| curtain | end table | plate | backpack | paper towel roll | basket | trash can | divider | toaster |
| blinds | dining table | soap dispenser | bag | book | bathroom cabinet | stair rail | oven | stool |
| shower curtain | keyboard | bucket | bed | bookshelf | bathroom counter | toaster oven | dish rack | plant |
| bookshelf | bag | clock | blanket | cart | bathroom stall | bulletin board | mini fridge | folded chair |
| tv | toilet paper | guitar | case of water bottles | furniture | bathroom stall door | dining table | bicycle | microwave |
| kitchen cabinet | printer | toilet paper holder | ceiling | blackboard | bathroom vanity | stuffed animal | laptop | cushion |
| pillow | blanket | speaker | closet | projector | bathtub | bathroom vanity | armchair | bench |
| lamp | microwave | cup | closet door | seat | bottle | box | couch | soap dispenser |
| dresser | shoe | paper towel roll | closet wall | folded chair | broom | ceiling | coffee kettle | storage organizer |
| monitor | computer tower | bar | clothes | office chair | clothes dryer | potted plant | counter | shower curtain |
| object | bottle | toaster | coat rack | projector screen | cushion | luggage | structure | cart |
| ceiling | bin | ironing board | container | whiteboard | doorframe | closet wall | pipe | kitchen counter |
| board | ottoman | soap dish | curtain | bin | fire alarm | paper cutter | bowl | towel |
| stove | bench | toilet paper dispenser | door | bucket | hair dryer | desk | shower curtain rod | blackboard |
| closet wall | basket | fire extinguisher | dresser | bulletin board | ledge | object | sofa chair | tv |
| couch | fan | ball | dumbbell | copier | light switch | rail | clothes dryer | printer |
| office chair | laptop | shower curtain rod | fan | machine | mat | shower head | coffee table | stand |
| kitchen counter | person | paper cutter | guitar case | mailbox | mirror | bed | stairs | rack |
| shower | paper towel dispenser | tray | hat | paper cutter | paper towel dispenser | paper towel dispenser | toilet seat cover dispenser | bathroom counter |
| closet | oven | toaster oven | ironing board | printer | plunger | fire extinguisher | machine | closet rod |
| doorframe | rack | mouse | lamp | column | scale | paper towel roll | paper bag | bottle |
| sofa chair | piano | toilet seat cover dispenser | laptop | storage container | shower | backpack | book | range hood |
| mailbox | suitcase | storage container | laundry basket | blinds | shower curtain | water bottle | blinds | purse |
| nightstand | rail | scale | laundry hamper | structure | shower curtain rod | bathroom cabinet | monitor | candle |
| washing machine | container | tissue box | luggage | water bottle | shower door | stove | shower wall | person |
| picture | telephone | light switch | mattress | ball | shower floor | laundry basket | curtain | coffee maker |
| book | stand | crate | mini fridge | board | shower head | alarm clock | closet | light switch |
| sink | light | power outlet | nightstand | box | shower wall | headphones | telephone | storage container |
| recycling bin | laundry basket | sign | object | cabinet | sink | piano | fan | bathroom stall door |
| table | pipe | projector | pillow | cd case | soap dish | guitar | ball | refrigerator |
| backpack | seat | candle | poster | ceiling light | soap dispenser | bag | bucket | fire alarm |
| shower wall | column | plunger | power outlet | clock | toilet | door | sign | tube |
| toilet | bicycle | stuffed animal | purse | computer tower | toilet paper | speaker | mirror | toilet paper holder |
| copier | ladder | headphones | rack | cup | toilet paper dispenser | water cooler | clock | ceiling light |
| counter | jacket | broom | recycling bin | desk | toilet paper holder | shoe | nightstand | picture |
| stool | storage bin | guitar case | shelf | divider | toilet seat cover dispenser | water pitcher | tv stand | end table |
| refrigerator | coffee maker | dustpan | shoe | file cabinet | towel | dumbbell | handicap bar | closet door |
| window | dishwasher | hair dryer | sign | headphones | trash bin | furniture | poster | file cabinet |
| file cabinet | machine | water bottle | storage bin | keyboard | washing machine | decoration | blanket | crate |
| chair | mat | handicap bar | storage organizer | monitor | closet rod | radiator | cup | toilet paper dispenser |
| plant | windowsill | purse | suitcase | mouse | dustpan | plunger | recycling bin | pillow |
| coffee table | bulletin board | shower floor | tissue box | paper | laundry detergent | shower | lamp | mat |
| stairs | fireplace | water pitcher | wardrobe | person | stuffed animal | bar | scale | bathroom stall |
| armchair | mini fridge | bowl | decoration | power strip | bowl | hair dryer | mouse | broom |
| cabinet | water cooler | paper bag | armchair | radiator | calendar | suitcase | wardrobe | container |
| bathroom vanity | shower door | alarm clock | bench | stand | coffee kettle | cabinet | ottoman | seat |
| bathroom stall | pillar | music stand | bicycle | telephone | coffee maker | chair | paper | jacket |
| mirror | ledge | laundry detergent | candle | tray | counter | board | power strip | dresser |
| blackboard | furniture | dumbbell | chair | tube | dish rack | laundry detergent | fireplace | dustpan |
| trash can | cart | tube | coffee table | window | dishwasher | whiteboard | doorframe | table |
| stair rail | decoration | cd case | couch | windowsill | fire extinguisher | vacuum cleaner | toilet | projector |
| box | closet door | closet rod | dining table | pipe | kitchen cabinet | power outlet | trash bin | window |
| towel | vacuum cleaner | coffee kettle | end table | stair rail | kitchen counter | storage bin | case of water bottles | windowsill |
| door | dish rack | shower head | fireplace | stairs | microwave | computer tower | light | tray |
| clothes | range hood | keyboard piano | jacket | | oven | mailbox | washing machine | cd case |
| whiteboard | projector screen | case of water bottles | light | | paper bag | shelf | guitar case | soap dish |
| bed | divider | folded chair | music stand | | plate | ledge | sink | office chair |
| bathtub | bathroom counter | fire alarm | ottoman | | range hood | pillar | bathtub | dishwasher |
| desk | laundry hamper | power strip | piano | | refrigerator | toilet paper | ladder | vent |
| wardrobe | bathroom stall door | calendar | picture | | stove | | bookshelf | coat rack |
| clothes dryer | ceiling light | poster | pillar | | toaster | | column | calendar |
| radiator | trash bin | luggage | plant | | toaster oven | | clothes | bin |
| shelf | bathroom cabinet | | potted plant | | trash can | | keyboard piano | projector screen |
| | calendar | | rail | | vent | | music stand | |
| | structure | | sofa chair | | water cooler | | | |
| | storage organizer | | speaker | | water pitcher | | | |
| | potted plant | | stool | | crate | | | |
| | mattress | | table | | ladder | | | |
| | | | tv | | | | | |
| | | | tv stand | | | | | |
| | | | vacuum cleaner | | | | | |

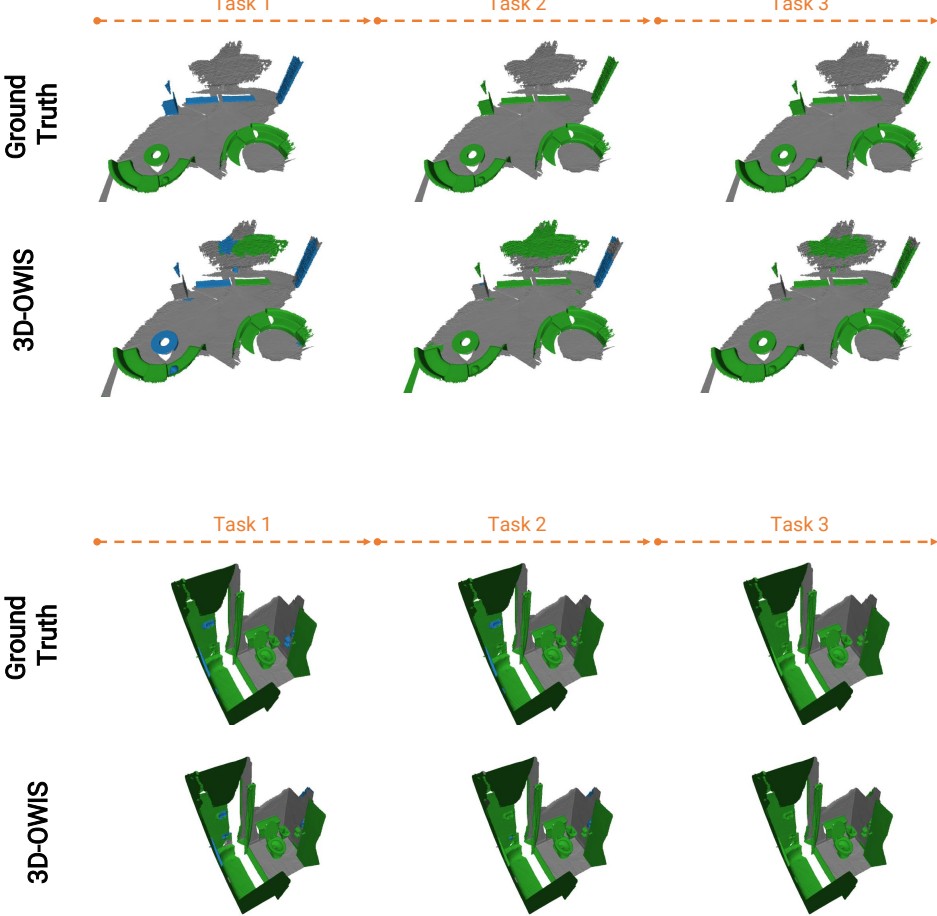

Figure 9: **Alleviating catastrophic forgetting during incremental learning.** The capability of **3D-OWIS** in retaining knowledge of the previously known classes after learning the new one is demonstrated across Task 2 and Task 3 for both scenes, where all objects of old known classes are still being predicted as known.

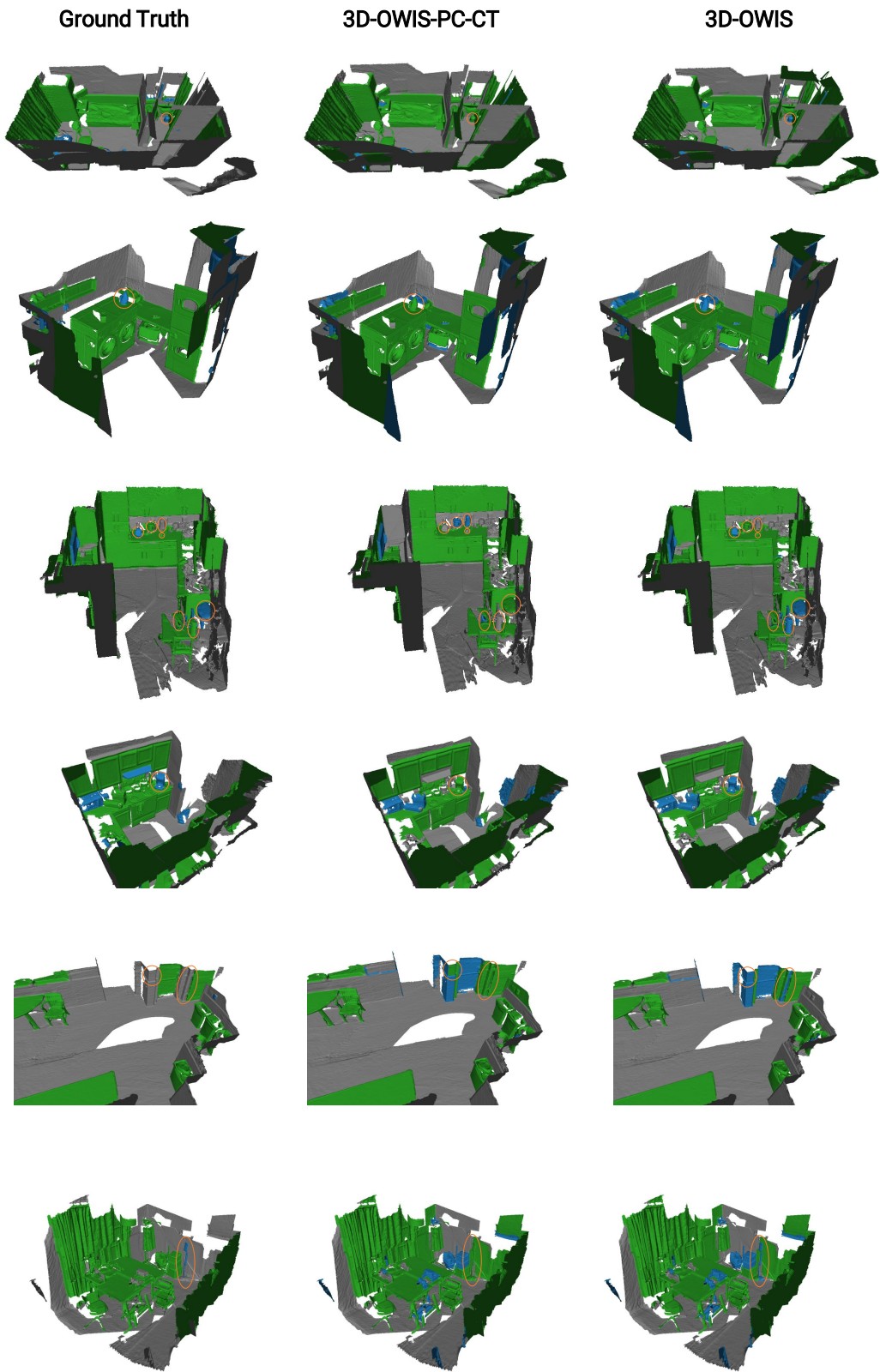

Figure 10: **Qualitative results.** The objects depicted in green represent the known classes, while the ones in blue represent the "unknown" class, and the gray objects represent the background. To emphasize the differences between **3D-OWIS** and **3D-OWIS−PC−CT**, we highlight them with orange circles.

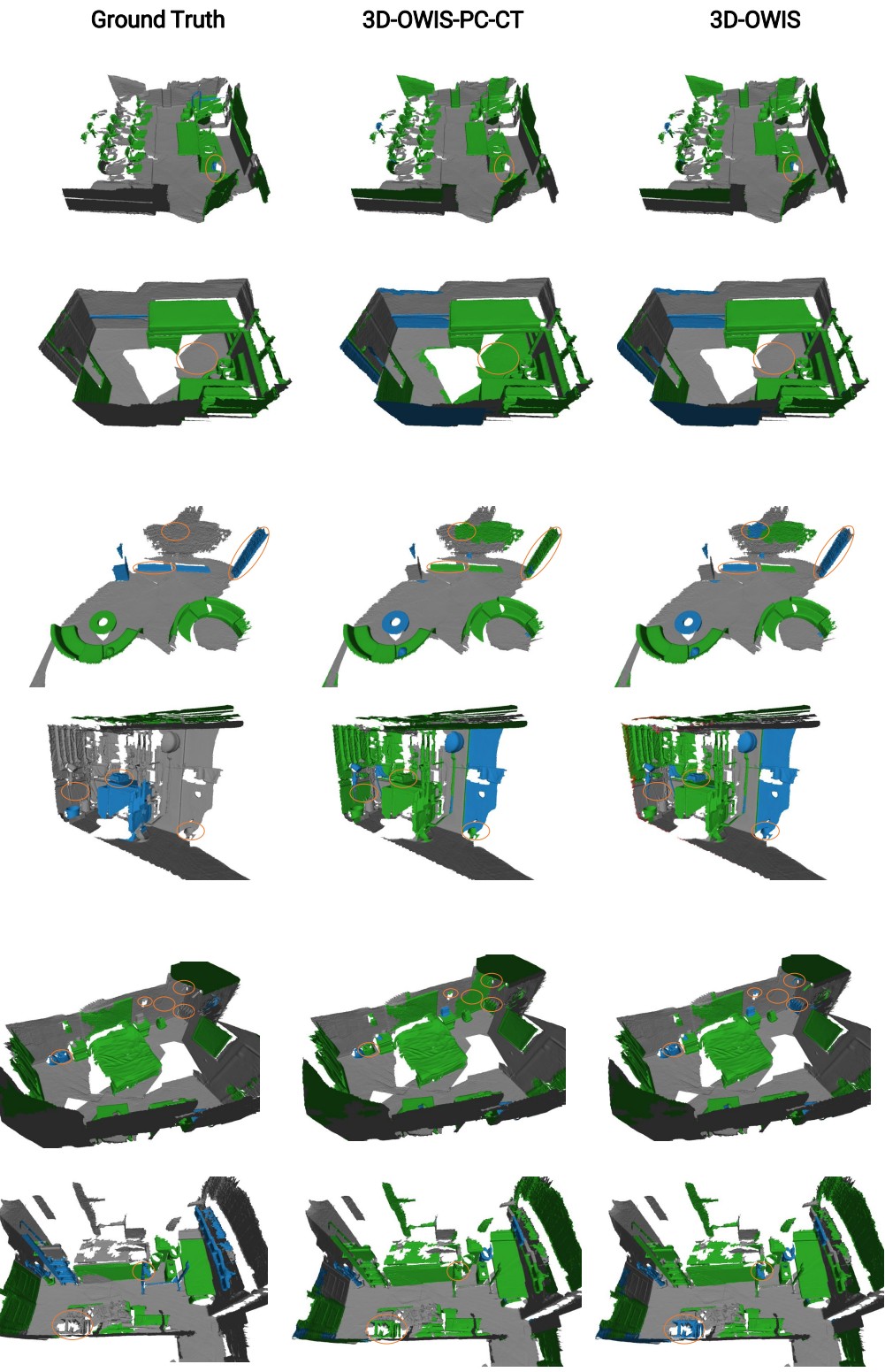

Figure 11: **Additional qualitative results** We demonstrate the better performance of our model in accurately identifying background objects (depicted in gray) as unknown (represented by the blue color), and also correcting the predictions from known class to unknown class. This capability greatly reduces the misclassification of background objects as known objects, leading to improved overall classification accuracy.

**Ground Truth**     **3D-OWIS-PC-CT**     **3D-OWIS**

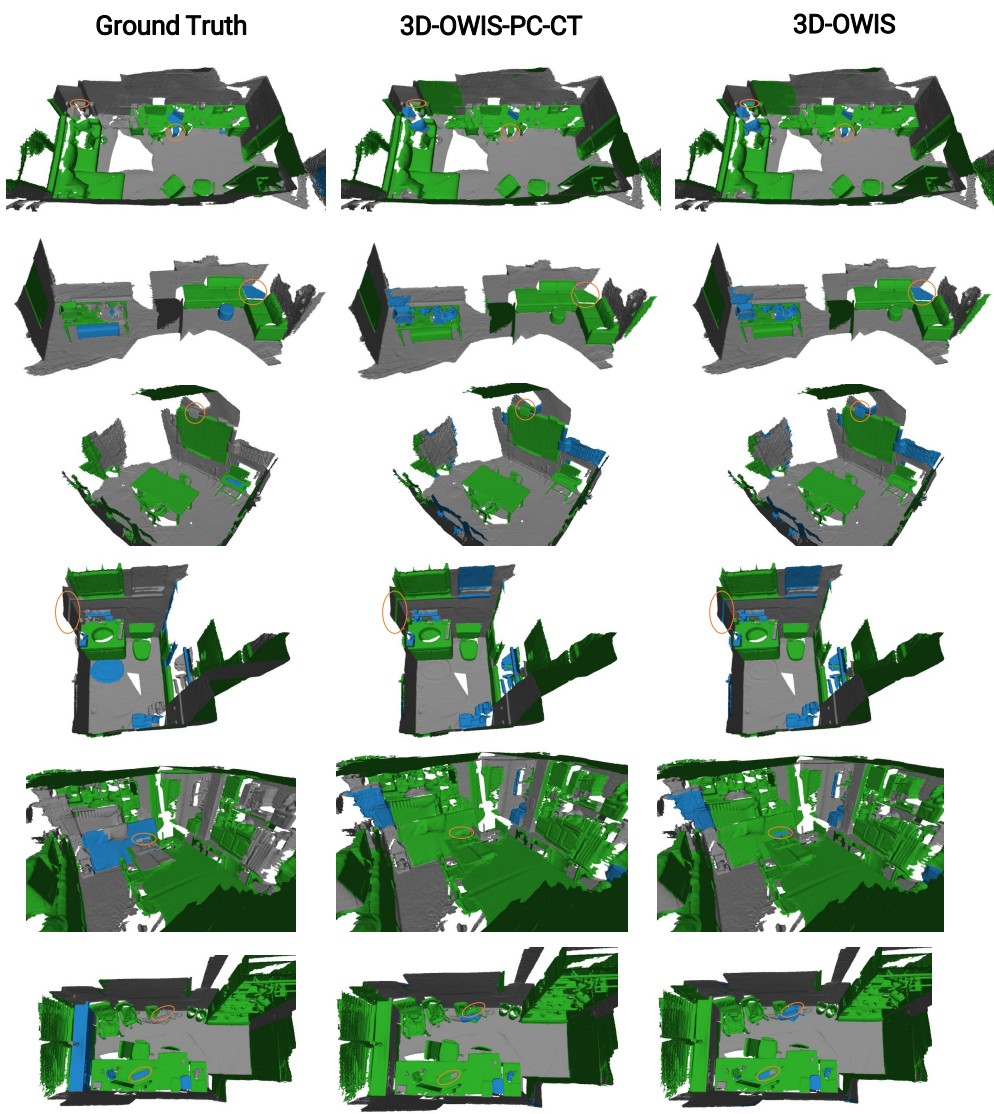

Figure 12: **Additional qualitative results**

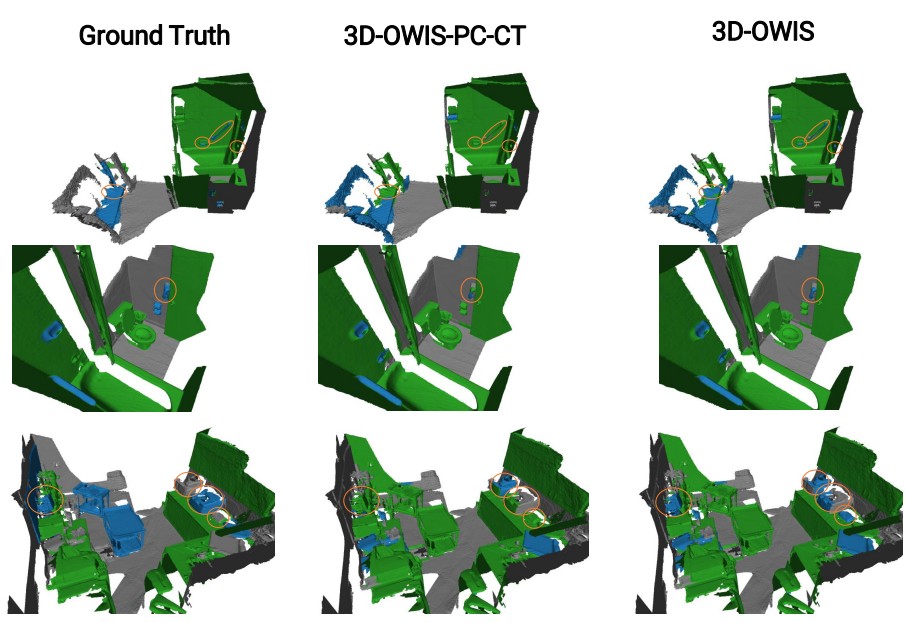

Figure 13: **Additional qualitative results**

