# OpenReview forum: "3D Indoor Instance Segmentation in an Open-World"
_NeurIPS.cc/2023/Conference — NeurIPS 2023 poster_

### Official Review · Reviewer_HHFW · 2023-07-01

**Soundness:** 2 fair
**Presentation:** 1 poor
**Contribution:** 2 fair
**Rating:** 5
**Confidence:** 2

**Summary:**

This work tackles the task of incremental object-discovery for 3D semantic instance segmentation. Unlike numerous concurrent works, it is not unsupervised but enables users (or oracle) to label objects that were identified as unknown in each iteration. The method is evaluated on ScanNet200, the paper proposes three new splits/tasks and measures three different metrics as defined in prior work.

**Strengths:**

The proposed approach has significant practical significance as it enables training on a labelled corpora of data and provides the option to users to label the identified unknown objects and then refine the trained model once training labels are provided by a user.


**Weaknesses:**

The manuscript is unclear on numerous occasions to the point that it is hard to understand the method (see questions below, these should be addressed in the rebuttal and incorporated in an updated version). In general, the paper might be clear for those that worked on this project (I assume for a significant amount of time) but the writing is challenging to understand by someone that reads about the project for the first time (unfortunately this makes it really difficult to write a meaningful review).

**Questions:**

Things that are not clear and should be improved:

- Looking at the videos and experiments: There are two methods 3D-OWIS-PC-CT and 3D-OWIS - are both yours? Or is one the baseline that is not yours, and the other yours? Maybe you can clearly mark yours with “Ours”.

- l.106-110: “the learned that updates itself” -- what does it mean? Is it correct that after one iteration of predictions by the model, the users labels the unknown classes, and then (in the next iteration) the model is trained again based in the existing and additionally provided user annotations? If this is the case, it is confusing to say “continuously improving itself (l.109)” since that somehow suggests that the model can improve without human intervention whereas here it seems to rely on human labels? I do not question the usefulness of user input after each iteration (i think it is very desirable), I simply do not understand what is happening in the end - the description should be improved.

- Figure 3 / Sec 3.2 Split B - I do not understand the split - what does it mean “exploring indoor areas” / “accessing an indoor space”? Is there a robot walking around in a simulation (I guess not)?

- l.132 - what is “the auto labeler”? The term is used here for the first time without explanation. The text then continues in l.136-143 explaining the drawbacks and proposed solutions. Since I don’t know what an auto-labeler is, these two paragraphs did not provide any information to me so I ignored them for writing this review. L. 144 “the auto labeller depicted in Fig.2” - it is not depicted in Fig2 - there is only a box with the name “auto labeled” - this does not provide any useful information to understand what is happening.

- Eq. 2, what is a class prototype? Do we have a class for each semantic class i.e. max of 200 in ScanNet200, or does it refer to individual object instances? The text suggests that it refers to semantic classes. If that is the case, how do you separate different instances that have the same semantic class?

- l.175 - what is 3D-OWIS? Is it the same as [20]? After reading l.172-177, they seem to be the same, but I could not find the name 3D-OWIS in [20]. Is 3D-OWIS maybe the name of the method proposed in this paper? The caption of  Fig.6 it seems to become clear that 3D-OWIS is the name of the method proposed in this paper. Similarly, in line 244 what is -PC-CT? What are the contributions? What does PC CT stand for? The last line of the caption in Tab.3 indicates that PC is probability corrections dn CT is confidence threshold? [I have to play detective only to understand what is the name of your method, why do you do this to me? Imagine R2 - they will directly reject the paper :( ]

- Why are the Tasks 1-2-3 introduced? To my understanding, it seems to be a redefinition of the existing head-common-tail classes. Why confuse the reader and introduce an alias for sth that is already clearly defined?


**Limitations:**

No / not applicable

---

> ### Author Rebuttal · Authors · 2023-08-09
>
> We thank the reviewer for the encouraging and insightful comments. Please find our responses to specific queries below.
>
> **Clarification on 3D-OWIS-PC-CT.**
> 3DOWIS-PC-CT is an extended Mask3D with predictive capabilities to encompass unknown classes. Unlike the original closed-set baseline Mask3D, which is confined to training on a predetermined set of classes and limited to predicting solely within those classes during inference, our novel approach leverages insights from ORE [1] to achieve open-world segmentation capabilities. We meticulously adapted and integrated these insights to 3D point cloud instance segmentation, where we employ the contrastive clustering on the final refined queries in Mask3D decoder to improve the objectness score estimation, which will help in increasing the quality of the generated unknown pseudo labels. An Auto-labeling module is used in 3DOWIS-PC-CT to select the predicted masks with the highest confidence (top-k) for instances that do not correspond to the ground truth of known classes. These predicted masks are then utilized as pseudo labels to represent the unknown class.
>
> **Continuous improvement.**
> In an open-world learning scenario, a model's progression is split into distinct tasks. In each task, the model is introduced to a specific group of labels that are known, while also being presented with numerous unfamiliar classes. In the first task, the model learns from the known examples and develops the ability to recognize unfamiliar classes within the scene. Moving on to the second task, the model builds on its knowledge from the known classes in the first task and is introduced to a fresh set of known classes. The concept of continuous improvement involves the model efficiently training on these solely new classes without causing its performance on the known classes from the first task to decline. A good model can retain the segmentation ability of old classes during training on new classes. This iterative process continues until the model reaches its maximum capacity in terms of the number of classes it can accommodate, determined by the classification head's size.
>
> **Split B motivation.**
> The perfect model for a robot moving indoors should segment both classes it knows and classes it hasn't seen before. Additionally, it should keep learning and getting better at segmenting new classes over time. In split B, we try to emulate how object classes might be sequentially labeled based on the scene types a robot first encounters. This grouping is good to assess how well models can segment objects in scenes that robots encounter while navigating in indoor areas.
>
> **Clarification on the auto-labeler.**
> A closed-set model's prediction only identifies classes it is familiar with, so all masks it predicts get labeled with one of those known classes. To create a substitute "pseudo-ground truth" for the unfamiliar classes, the auto-labeler picks predicted masks with the best objectness and zero overlaps with the known class's actual mask. These picked masks are marked as unknown and help teach the model about the unknown cases.
>
> **Clarification on the class prototype.**
> A class prototype is a summary feature that captures both the spatial and semantic aspects of a class. If, for example, there are 20 known classes, the model stores 21 of these summaries: one for each known class and an extra one for the unknown classes. These prototypes are estimated by keeping an exponential moving average of the features matched with the ground truth for the known classes and the pseudo-ground truth for the unknown classes.
>
> Note that in Mask3D 100 queries are refined and each query is used to generate a mask and a class, but only some of them correspond to a real ground truth. As a result, the 100 masks and labels generated from the queries are matched with the ground truth for the knowns and pseudo-ground truth for the unknowns using Hungarian matching, then the query feature is selected to represent its matched ground truth, and then stored to perform the class prototype estimation of the classes using exponential moving average.
>
> **Clarification on the 3D-OWIS and 3D-OWIS-PC-CT.**
> As discussed above, 3D-OWIS-PC-CT represents our refined adaptation of the initial closed-set baseline Mask3D. The goal here was to enhance its capabilities for an open-world environment. On the other hand 3D-OWIS is our final model, which emerges from two key changes:
>
> - Firstly, we replaced the selection of top-k-based unknown pseudo labels selection with Confidence Threshold (CT) -based selection. This change, which prioritizes quality over quantity, contributes to generating improved masks and enhancing performance on known classes.
>
> - Secondly, we integrated Probability Correction (PC), a technique aimed at rectifying instances that were initially misclassified as known classes. Our inspiration for PC came from analyzing t-SNE plots of the features of unknown classes, a comparison that revealed greater dispersion in our unknown class features compared to those of ORE [1] in the 2D open-world object detection (see **rebuttal pdf**). This observation shows the challenge of segmenting unknowns in a 3D setting as opposed to 2D.
>
> **Need for Tasks 1, 2, and 3.**
> A task refers to what set of classes the model has to learn. It is used to keep track of the progress of a model while learning new classes. for split A, which is based on class-frequency, it follows the head/common/tail splitting in ScanNet200. However, split B and C are distinct from this classification in the original dataset, where split B is determined based on regions and scene types and split C is completely random to simulate the randomness aspect of the open-world.
>
> [1] Joseph, K. J., et al. Towards open world object detection, CVPR 2021.

---

> > ### Comment · Reviewer_HHFW · 2023-08-15
> > **Thanks for the rebuttal!**
> >
> > I thank the authors for the extensive rebuttal which provided insights to my remaining questions. An updated version can be improved by including those clarifications as well as the additional experiments.

---

### Official Review · Reviewer_L857 · 2023-07-04

**Soundness:** 2 fair
**Presentation:** 2 fair
**Contribution:** 3 good
**Rating:** 5
**Confidence:** 3

**Summary:**

This paper proposes a pipeline to do 3D open-world instance segmentation. The authors provide a problem definition and introduced three different scenarios. Moreover, to overcome the possible problems that may lead to lower performance on known classes, the authors proposes different modules like probability correction and exemplar replay. The comprehensive experiments show their proposed methods' effectiveness.

**Strengths:**

As claimed by the author, this paper is the first to investigate the 3D open-world instance segmentation task, and develop some well-motivated modules to improve the performance upon the simple baseline.

**Weaknesses:**

I'm not very familiar with the open-world related task and common practices both in 2D and 3D. Currently I think the writing of this paper is confusing and seems to be finished in a hurry. For example, the "3D-OWIS-PC-CT", I assume the "-PC-CT" is **without PC and CT**? And I couldn't find explanations about the "PC" and "CT".

In addition, I think authors can also provide the results of their pipeline equipped with other 3D instance segmentation as a comparison.

**Questions:**

As mentioned before I'm not an expert in this field, I'll ask more questions in later reviewing process. Currently I have one concern, since the network won't be re-trained in test and the network structure is fixed, does it mean the total class number (including the known classes and the unknown classes) will have an upper bound? And will the performance of this feature-cluster-based new class discovery become lower as the new class number increases?

**Limitations:**

No. The authors didn't address their limitations, and also the possible negative societal impact. I would suggest authors discuss some foundamental drawbacks of current open-world discovery settings.

---

> ### Author Rebuttal · Authors · 2023-08-09
>
> We thank the reviewer for the encouraging and insightful comments. Please find our responses to specific queries below.
>
> **Explanation about PC and CT.**
> 3D-OWIS-PC-CT represents our method without Probability Correction (PC) and without Confidence Threshold (CT). The final model 3D-OWIS includes all components, including PC and CT. Our 3D-OWIS replaces the selection of top-k-based unknown pseudo labels selection with Confidence Threshold (CT) -based selection. This change, which prioritizes quality over quantity, contributes to generating improved masks and enhancing performance on known classes. Further, we integrated Probability Correction (PC), a technique aimed at rectifying instances that were initially misclassified as known classes. Our inspiration for PC came from analyzing t-SNE plots of the features of unknown classes, a comparison that revealed greater dispersion in our unknown class features compared to those of ORE [1] in the 2D open-world object detection (see rebuttal pdf). This observation shows the challenge of segmenting unknowns in a 3D setting as opposed to 2D.
>
> **Comparison to other 3D instance segmentation.**
> We base our work on the recently introduced Mask3D (ICRA 2023), as it achieves state-of-the-art performance on the challenging ScanNet200 benchmark for the task of 3D indoor instance segmentation. Our approach on three carefully curated open-world splits achieves promising results compared to the recent Mask3D. As a potential future work, we aim to extend our open-world setting for outdoor 3D scenes and approaches.
>
> **Bound on the number of classes.**
> In our experiments, we set the maximum number of classes to 200, which includes all the classes provided by ScanNet200 benchmark. Nonetheless, increasing the limit of the number of classes would only add more classifiers at the last layer of Mask3D. These classifiers are only trained once the network is presented with ground-truth objects of corresponding classes, and do not affect the overall performance of previous tasks. Therefore, increasing the limit of the number of classes would yield similar results to the ones provided in the paper, but with additional memory requirements.
>
> | \# of classes   | 200   | 1000  | 5000  | 10000 | 50000 | 100000 |
> | --------------- | ----- | ----- | ----- | ----- | ----- | ------ |
> | Size of 3D-OWIS | 39.7M | 39.8M | 40.7M | 41.9M | 50.9M | 62.2M  |
>
> **Discussion of limitations.**
> We will add the limitations and societal impact in the revised version. Similar to our base architecture Mask3D, we assume that complete reconstructed scenes are used as input, so the learning benefits from contextual information.
>
> [1] Joseph, K. J., et al. Towards open world object detection, CVPR 2021.

---

> > ### Comment · Reviewer_L857 · 2023-08-14
> >
> > After reading the authors' response, I think most of my concerns are solved. As I'm not an expert in this field, I have also read other reviews and think authors provide useful feedbacks towards them. I will keep my ratings as borderline accept or weak accept, and would love to see other reviewers' post-rebuttal comments to make final decisions.

---

### Official Review · Reviewer_WNwG · 2023-07-05

**Soundness:** 3 good
**Presentation:** 3 good
**Contribution:** 2 fair
**Rating:** 5
**Confidence:** 4

**Summary:**

This paper introduces an open-world 3D indoor instance segmentation method, where an auto-labeling scheme is employed to produce pseudo-labels during training and induce separation to separate known and unknown category labels. Pseudo label quality are further improved subsequently.

**Strengths:**

1. Open world 3d instance segmentation is an important problem and the author first addresses this problem.
2. The method designing is reasonable and complete.
3. The experiment is also sufficient.

**Weaknesses:**

1. Some designs are hard to understand. For example, for probability correction, why unknown object has to be far from the nearest known class? since we have no prior about category distributions. In experiments, you can also see that PC does not bring improvement to all metrics. This should be illustrated more clearly.
2. Although such open world task has not been studied in 3d tasks, it has been widely studied in 2d tasks. Since the core issue of this problem is actually the same, the author should implement some 2d open world methods in 3d instance segmentation and compare with them.

**Questions:**

Please refer to the weakness section.

**Limitations:**

The author does not discuss the limitation.

---

> ### Author Rebuttal · Authors · 2023-08-09
>
> We thank the reviewer for the suggestions to improve the clarity of the paper.
>
> **On probability correction assuming unknowns are far from knowns.**
> In our open-world 3D instance segmentation framework, we use the auto-labeler to generate pseudo-labels for the unknowns. The contrastive clustering step then minimizes the distance between the queries and the corresponding class prototype and maximizes the distance between the other class prototypes. This includes queries corresponding to the pseudo-labels that are pulled toward the unknown class prototype. Since the class prototypes are pushed away from each other, it is expected that the unknowns are also pushed away from the known class prototypes. We thank the reviewer and will add this clarification in the revised version.
>
> **Comparison to 2D baselines.**
> We empirically observe that a direct integration of different strategies from 2D provides sub-optimal results with respect to the trade-off between unknown recall and average AP for the proposed open-world 3D indoor instance segmentation setting. To this end, we conduct experiments by leveraging contrastive cluster (ORE [4]) or attention-driven pseudo-labeling (OW-DETR [3]) from recent 2D detection works. These experiments and their results are presented below:
>
> - OW-DETR [3] has introduced an objectness head to predict the instances' objectness in an input image, which significantly improves the performance of the 2D model for both known and unknown classes. However, our experiments revealed that learning the objectness of 3D features enhances the prediction recall for unknown classes (similar to 2D models) but adversely affects the performance of the 3D model on known classes (Ours with learnable objectness as in OW-DETR: 33.35 mAP vs: **Ours: 39.70** mAP).
>
> - Given the enriched semantic and 3D spatial information in the refined queries of the transformer decoder in Mask3D, we pursued a contrastive clustering approach to enhance the distinctiveness between queries of different classes. This facilitates a more accurate estimation of the objectness for predicted instances. To cater to the 3D nature of the problem, we tailored the clustering technique used in ORE [4] for OWOD. Unlike OWOD which clusters intermediate features in the detector, we choose to cluster the queries that encapsulate the richer spatial and semantic information extracted from the 3D transformer decoder rather than features from the output of the 3D backbone.
>
> Moreover, to gain more insight into the behavior of unknown classes in the 3D setting, we visualize the t-SNE plots of class features used for clustering in ORE [4] for 2D images and in our 3D-OWIS for 3D point clouds. The t-SNE comparison is presented in the **rebuttal pdf**. The t-SNE features in ORE [4] are extracted from the detector where Pascal VOC classes are known and all classes from MS-COCO are grouped together as unknown. In the case of our 3D-OWIS for 3D point clouds, we use the final refined queries in Mask3D decoder, which are used for predicting the masks and the class labels. In our 3D setting, we are showing split A task 1. The results in the figure show more sparsity in the 3D features of the unknown classes which makes them harder to segment compared to their 2D counterparts which are easier to cluster. To cope with this challenge in 3D point cloud unknown segmentation, we propose to use the known class query prototypes to correct the features of the unknowns in the boundaries of the clusters of the known classes.
>
> Therefore, we propose to tailor our method to these observations in the 3D domain and aim to provide an optimal trade-off between unknown recall and known AP. The proposed scheme provides superior performance compared to 2D strategies as shown in the table below. Further, we also present a comparison with two other existing works OLN [1] and GGN [2] in the table below. For OLN [1] implementation, we remove the classification head. In the case of GGN [2], we train an affinity predictor with Minkowski 3D backbone. We hypothesize that OLN failed in 3D point cloud instance segmentation because it wasn't able to learn a good objectness representation from masks only given the sparsity of point clouds. Meanwhile, GGN [2] pseudo-labels are of lower quality since the affinity is not optimal for 3D because of the empty space and the disconnected object parts (qualitative results in **rebuttal pdf** show GGN pseudo labels).
>
> |                       |             | WI ↓    | A_OSE ↓| U-Recall ↑| mAP(known)  ↑    | mAP ↑  |
> |-------------------|-----------|:---------:|:-----------:|:--------------:|:----------------------:|:----------:|
> | Closed-set    | Oracle  | 0.129 | 227   | 55.94    | 38.75                     | 38.6  |
> |                      | Mask3D  | \-      | \-    | \-    | 39.12    | 39.12                     |
> |||||||
> | Open World  | OW-DETR [3] | 0.547 | 721   | 22.14    | 35.56                     | 35.05 |
> |                      |  GGN  [2]                 | 15.68      | 1452    | 21.33    | 20.51      | 20.12                        |
> |                      |  OLN   [1]            | \-      | \-    | 02.45  | \-       | \-                        |
> |                      | **3D-OWIS (Ours)**              | 0.397   | 607   | 34.75 | 40.2     | 39.7                      |
>
>
> **Discussion of limitations.**
> We will add the limitations and societal impact in the revised version. Similar to our base architecture Mask3D, we assume that complete reconstructed scenes are used as input, so the learning benefits from contextual information.
>
> [1] Kim et al., Learning Open-World Object Proposals without Learning to Classify, ICRA 2022
>
> [2] Wang et al., Open-World Instance Segmentation: Exploiting Pseudo Ground Truth From Learned Pairwise Affinity, CVPR 2022
>
> [3] Gupta, Akshita, et al. Ow-detr: Open-world detection transformer, CVPR 2022.
>
> [4] Joseph, K. J., et al. Towards open world object detection, CVPR 2021.

---

> > ### Comment · Reviewer_WNwG · 2023-08-16
> > **Thanks for the rebuttal**
> >
> > Thank you for your rebuttal. Most of my concerns are addressed, and I will keep my score.

---

### Official Review · Reviewer_CNBM · 2023-07-05

**Soundness:** 3 good
**Presentation:** 2 fair
**Contribution:** 2 fair
**Rating:** 6
**Confidence:** 3

**Summary:**

The paper presents a new application of open-world object detection to the setting of 3D instance segmentation. In this setup, a 3D point-cloud segmentation model is required to label each point with instance information, whether this instance is part of training or not. The paper uses ScanNet and proposes a few ways to partition the dataset to use for evaluation in open-world. Finally, the paper adopts the framework in [16] as a baseline for this problem and proposes two tricks on top this framework: PC (probability correction) and CT (confidence thresholding).

**Strengths:**

The task of open-world segmentation in 3D is important yet overlooked by previous work. As outlined by the paper, prior works mainly focus on 2D setups, either image or video. The use of ScanNet for evaluation seems appropriate and the splitting mechanism also seems reasonable, especially the Region-Based split has a very nice motivation from real-world application.


**Weaknesses:**

Post rebuttal comments:
1. The author provided more insights towards the difference between 2D and 3D. I was hoping for more attributes unique to 3D. But having some insights as least improve the quality of the work.
2. My second concern is addressed.


Original review:
Although there are many things to like about the proposal of the task, this work may suffer from a few important weaknesses:
- Missing insights specifically to 3D. The major contribution of the paper is to adopt the recent new task of open-world localization to 3D setting. Unfortunately, the paper does not focus enough on aspects unique to 3D. Most components are the same as the 2D task and insights provided by the paper do not contain enough 3D-specific information. To justify the contribution, it is important for the paper to provides both intuitions and empirical proves on why it is worth studying 3D and how it is different than simple adoption of a 2D framework. In fact, the 3D-OWIS seems to be almost identical to [16] besides swapping the backbone predictor with a 3D backbone. The claimed contributions PC and CT are also not specific to 3D or reveals anything special about 3D.
- Lack of comprehensive evaluation/ ablations. If author would prefer to claim more of their contributions on the dataset part, it is important to benchmark the task comprehensively. A suite of recent baselines in 2D should be evaluated. For example, recent works OLN [A] and GGN [B] both provide good class-agnostic proposals, and their 3D variants should be evaluated. In particular, GGN also uses a self-teaching schema.

Minor:
- The presentation of baseline 2D-OWIS-PC-CT seems really vague when first-time introduced. Better elaborate on what this is when it is discussed the first time.
- If author prefers to claim contributions mainly to the task/ benchmark, NeurIPS D&B is more appropriate.

[A] Kim et al., Learning Open-World Object Proposals without Learning to Classify, ICRA 2022
[B] Wang et al., Open-World Instance Segmentation: Exploiting Pseudo Ground Truth From Learned Pairwise Affinity, CVPR 2022

**Questions:**

NA

**Limitations:**

Limitation is not sufficiently discussed. For example, when repurposing ScanNet200, what are some potential limitation? This dataset isn't designed specifically for this proposed task in open-world.

---

> ### Author Rebuttal · Authors · 2023-08-09
>
> We appreciate the reviewer's valuable feedback. Detailed answers to the reviewer's queries are provided below.
>
> **3D-specific insights.**
> As recommended, we present here both empirical and qualitative results to provide more insights specific to our 3D setting. We empirically observe that a direct integration of different strategies from 2D provides sub-optimal results with respect to the trade-off between unknown recall and average AP for the proposed open-world 3D indoor instance segmentation setting. To this end, we conduct experiments by leveraging contrastive cluster (ORE [16]) or attention-driven pseudo-labeling (OW-DETR [C]) from recent 2D detection works. These experiments and their results are presented below:
>
> OW-DETR [C] has introduced an objectness head to predict the instances' objectness in an input image, which significantly improves the performance of the 2D model for both known and unknown classes. However, our experiments revealed that learning the objectness of 3D features enhances the prediction recall for unknown classes (similar to 2D models) but adversely affects the performance of the 3D model on known classes (Ours with learnable objectness as in OW-DETR: 33.35 mAP vs: **Ours: 39.70** mAP).
>
> Given the enriched semantic and 3D spatial information in the refined queries of the transformer decoder in Mask3D, we pursued a contrastive clustering approach to enhance the distinctiveness between queries of different classes. This facilitates a more accurate estimation of the objectness for predicted instances. To cater to the 3D nature of the problem, we tailored the clustering technique used in ORE [16] for OWOD.  Unlike OWOD in [16] which clusters intermediate features in the detector, we propose to cluster the queries that encapsulate the richer spatial and semantic information extracted from the 3D transformer decoder rather than features from the output of the 3D backbone.
>
> Moreover, to gain more insight into the behavior of unknown classes in the 3D setting, we visualize the t-SNE plots of class features used for clustering in ORE [16] for 2D images and in our 3D-OWIS for 3D point clouds. The t-SNE comparison is presented in the **rebuttal pdf**. The t-SNE features in ORE [16] are extracted from the detector where Pascal VOC classes are known and all classes from MS-COCO are grouped together as unknown. In the case of our 3D-OWIS for 3D point clouds, we use the final refined queries in Mask3D decoder. In our 3D setting, we are showing split A task 1. The t-SNE results show more sparsity in the 3D features of the unknown classes which makes them harder to segment, compared to their 2D counterparts which are easier to cluster. To cope with this challenge in 3D point cloud unknown segmentation, we propose to use the known class query prototypes to correct the features of the unknowns in the boundaries of the clusters of the known classes.
>
> The proposed scheme provides superior performance compared to both 2D strategies as shown in the table below.
>
> **Comparison to OLN [A] and GGN [B].**
> As recommended, we present the comparison with OLN [A] and GGN [B] in the table below. For OLN [A], we remove the classification head from Mask3D, while we keep the mask loss during training. In the case of  GGN [B], we train an affinity predictor with Minkowski as a 3D backbone and a prediction head for the affinity.  Affinity maps are generated by the affinity predictor, and fed to the Connected Components module to generate class-agnostic mask proposals. We hypothesize that OLN failed in 3D point cloud instance segmentation because it wasn't able to learn a good objectness representation from masks only given the sparsity of point clouds. Meanwhile, GGN [B] pseudo-labels are of lower quality since the affinity is not optimal for 3D because of the empty space and the disconnected object parts (qualitative results in **rebuttal pdf** show GGN pseudo labels).
>
> |                       |             | WI ↓    | A_OSE ↓| U-Recall ↑| mAP(known)  ↑    | mAP ↑  |
> |-------------------|-----------|:---------:|:-----------:|:--------------:|:----------------------:|:----------:|
> | Closed-set    | Oracle  | 0.129 | 227   | 55.94    | 38.75                     | 38.6  |
> |                      | Mask3D  | \-      | \-    | \-    | 39.12    | 39.12                     |
> |||||||
> | Open World  | OW-DETR [C] | 0.547 | 721   | 22.14    | 35.56                     | 35.05 |
> |                      |  GGN  [B]                 | 15.68      | 1452    | 21.33    | 20.51      | 20.12                        |
> |                      |  OLN   [A]            | \-      | \-    | 02.45  | \-       | \-                        |
> |                      | **3D-OWIS (Ours)**              | 0.397   | 607   | 34.75 | 40.2     | 39.7                      |
>
> **Clarification on 3D-OWIS-PC-CT.**
> We thank the reviewer for the suggestion. 3D-OWIS-PC-CT represents our method without Probability Correction (PC) and without Confidence Threshold (CT). The final model 3D-OWIS includes all components, including PC and CT.
>
> **Discussion of limitations.**
> We note that ScanNet200 benchmark is the largest 3D indoor instance segmentation dataset in terms of the number of classes. Due to the highly challenging nature, diversity, and number of classes, we adapt it to open-world setting. We also introduce carefully curated open-world splits leveraging realistic scenarios based on inherent object distribution, region-based indoor scene exploration, and the randomness aspect of open-world classes. As noted by the reviewer, our region-based split is well motivated from the perspective of real-world application. In future, we aim to further explore open-world setting for outdoor 3D scenes.
>
> [C] Gupta, Akshita, et al. Ow-detr: Open-world detection transformer, CVPR 2022.

---

> > ### Comment · Reviewer_CNBM · 2023-08-11
> > **Thanks for your response**
> >
> > After reading the response from authors as well as reviews from other reviewers, I think my initial concerns were shared among other reviewers. Given the response, I think my second concern regarding lack of studies adopting 2D detectors is resolved. It seems non-trivial to extend them to 3D setting, and an naive extension with GGN fails due to the reasons explained by the authors.
> >
> > The author gave some analysis regarding 3D features being more scattered and difficult, and I think this insight is useful. However, I think what (other reviewers and) I are looking for, is something unique to 3D, such as geometry. For example, the failed adoption of OLN due to sparsity of point-cloud is quite interesting. Some other angles relating to 3D geometry might provide extra insights to strengthen the paper. For example, is 3D shape harder to generalize in open-world than 2D? Or maybe shapes are less prone to overfitting in closed-world scenario.
> >
> > That being said, I think the author did a good job responding to the concern regarding adopting other 2D detectors. I am inclined to raise my rating to borderline accept or weak accept. Would love to hear from other reviewers' post-rebuttal comments.

---

### Official Review · Reviewer_PXM7 · 2023-07-06

**Soundness:** 3 good
**Presentation:** 2 fair
**Contribution:** 3 good
**Rating:** 5
**Confidence:** 4

**Summary:**

This paper addresses the challenge of 3D instance segmentation in open-world scenarios.  It starts with a formulation for this problem, including the definition and setup of known and unknown objects and different splits of categories for simulating different open-world cases. Accordingly, this work proposes a framework incorporating an unknown object identifier to detect objects not present in the training set and devises several strategies to enhance the separation of classes within the query embedding space, such as contrastive clustering and reachability-based probability correction. Experiments on the new benchmark validate the effectiveness of the proposed framework.

**Strengths:**

- This paper studies a new problem setting, open-world 3D instance segmentation, with a new benchmark and framework.
- The basic idea is easy to follow and the presentation is overall clear.
- The illustration figures are also clear such as Fig. 2-4.
- The new benchmark considers different open-world cases when splitting different categories.
- The corresponding modifications tailored to this new problem are reasonable, from contrastive clustering for queries to reachability-based probability correction and alleviating catastrophic forgetting for incremental learning.
- Experiments show the effectiveness of the proposed framework. Different evaluation metrics also show different perspectives for comparing different methods.

**Weaknesses:**

- There is no preliminary section for the baseline framework, Mask3D, making the introduction of corresponding modifications tailored to this new setting a little confusing. In my opinion, it would be much better to introduce the baseline framework after setting up the benchmark after Sec. 3.2 and even open a new section for the "approach" part.

- Some of the modifications are adapted from experiences in the 2D domain, and they are not revised to fit the 3D problem or tackle some specific challenges in 3D. Although some of them such as reachability-based probability correction have their own insight, it is still unclear whether the underlying reason for these challenges is related to the different data modality. It would be better to have deeper thinking from this aspect.

- The evaluation benchmark involves comprehensive metrics for the new setup, but it lacks an intuitive connection or comparison between the new problem and other conventional ones, such as 2D open-world instance segmentation/detection and 3D close-set instance segmentation. Given this is a relatively new problem, it would be better to compare it with other well-explored problems to make it more friendly and acceptable for researchers from this community.

**Questions:**

None.

**Limitations:**

The author does not discuss the limitations and potential social impacts in the paper.

---

> ### Author Rebuttal · Authors · 2023-08-09
>
> We thank the reviewer for the encouraging and insightful comments. Please find our responses to specific queries below.
>
> **Preliminary section after Sec 3.2.**
> We thank the reviewer for this suggestion. As suggested by the reviewer, we will add a new section to describe our closed-set baseline “Mask3D” in the revised version.
>
> **More insights on the underlying 3D challenges.**
> The open-world learning problem requires detecting unknowns (unknown recall) along with accurately detecting class-specific knowns (average AP). While some of the proposed adaptations are inspired from the 2D domain, we note that a direct integration of these strategies from 2D provides sub-optimal results with respect to the trade-off between unknown recall and average AP for open-world 3D indoor instance segmentation. To gain further insights, we conduct experiments by leveraging contrastive cluster (ORE) [4] or attention-driven pseudo-labeling (OW-DETR) [3] from recent 2D detection works. These experiments and their results are presented below:
> - OW-DETR [3] has introduced an objectness head to predict the instances' objectness in an input image, which significantly improves the performance of the 2D model for both known and unknown classes. However, our experiments revealed that learning the objectness of 3D features enhances the prediction recall for unknown classes (similar to 2D models) but adversely affects the performance of the 3D model on known classes (Ours with learnable objectness as in OW-DETR: 33.35 mAP vs: **Ours: 39.70** mAP).
> - Given the enriched semantic and 3D spatial information in the refined queries of the transformer decoder in Mask3D, we pursued a contrastive clustering approach to enhance the distinctiveness between queries of different classes. This facilitates a more accurate estimation of the objectness for predicted instances. To cater to the 3D nature of the problem, we tailored the clustering technique used in ORE [4] for OWOD. Unlike OWOD in [4] which clusters intermediate features in the detector, we propose to cluster the queries that encapsulate the richer spatial and semantic information extracted from the 3D transformer decoder rather than features from the output of the 3D backbone.
>
> Moreover, we observed from the t-SNE plots of the known and unknown features (**provided in the rebuttal pdf**) that the features of the unknowns are more dispersed in 3D compared to 2D, and are thus harder to segment.
>
> Therefore, we propose to tailor our method to these observations in the 3D domain and aim to provide an optimal trade-off between unknown recall and known AP. The proposed scheme provides superior performance compared to 2D strategies as shown in the table below.
>
> **Comparison to 2D open-world instance segmentation/detection and 3D closed-set instance segmentation**.
> As recommended by the reviewer, we provide the following results after implementing techniques proposed for 2D open-world instance segmentation/detection. We also present a comparison between the results of Mask3D and Oracle (3D-OWIS with access to the training dataset of the previously known classes and the labels of the unknown classes). We show results for Split A task1, as we note that GGN [2] and OLN [1] focus on segmenting the unknowns and do not target the incrementally learned tasks. On the other hand, our method aims at both segmenting the unknowns in the current task, as well as incrementally learning classes that are introduced at each task. Our approach performs favorably compared to [1, 2, 3].
>
> |                       |             | WI ↓    | A_OSE ↓| U-Recall ↑| mAP(known)  ↑    | mAP ↑  |
> |-------------------|-----------|:---------:|:-----------:|:--------------:|:----------------------:|:----------:|
> | Closed-set    | Oracle  | 0.129 | 227   | 55.94    | 38.75                     | 38.6  |
> |                      | Mask3D  | \-      | \-    | \-    | 39.12    | 39.12                     |
> |||||||
> | Open World  | OW-DETR [3] | 0.547 | 721   | 22.14    | 35.56                     | 35.05 |
> |                      |  GGN  [2]                 | 15.68      | 1452    | 21.33    | 20.51      | 20.12                        |
> |                      |  OLN   [1]            | \-      | \-    | 02.45  | \-       | \-                        |
> |                      | **3D-OWIS (Ours)**              | 0.397   | 607   | 34.75 | 40.2     | 39.7                      |
>
>
> **On limitations**. We will add the limitations and societal impact in the revised version. Similar to our base architecture Mask3D, we assume that complete reconstructed scenes are used as input, so the learning benefits from contextual information.
>
> [1] Kim et al., Learning Open-World Object Proposals without Learning to Classify, ICRA 2022
>
> [2] Wang et al., Open-World Instance Segmentation: Exploiting Pseudo Ground Truth From Learned Pairwise Affinity, CVPR 2022
>
> [3] Gupta, Akshita, et al. Ow-detr: Open-world detection transformer, CVPR 2022.
>
> [4] Joseph, K. J., et al. Towards open world object detection, CVPR 2021.

---

> > ### Comment · Reviewer_PXM7 · 2023-08-15
> > **Response to Rebuttal**
> >
> > I acknowledge that I have read the authors' rebuttal and the other reviews.
> >
> > Thank you for addressing my concerns. I think most of my concerns are addressed to some extent and will keep my score in the final decision. I strongly recommend the author carefully consider the reviewers' comments and revise the paper as promised in the rebuttal.

---

### Author Rebuttal · Authors · 2023-08-09

We thank all the reviewers (PXM7, CNBM, WNwG, L857, HHFW) for the positive and valuable feedback, and we appreciate the comments to improve our work. **Reviewer PXM7:** "Idea is easy to follow and the presentation is overall clear. Illustration figures are clear. The corresponding modifications tailored to this new problem are reasonable". **Reviewer CNBM:**  "open-world segmentation in 3D is important yet overlooked. Region-Based split has a very nice motivation from real-world application". **Reviewer WNwG:** "The method designing is reasonable and complete. The experiment is also sufficient.". **Reviewer L857:** "develop some well-motivated modules to improve the performance". **Reviewer HHFW:** "The proposed approach has significant practical significance".

We summarize the main points presented in our response:
- We include comparisons to other strategies tackling 2D open-world.
- We present more insights on the 3D challenges and motivations to our proposed method.
- We provide clarifications and answers to the points raised, which we will include in our revised version.

---

### Decision · Program_Chairs · 2023-09-21

**Decision:**

Accept (poster)

**Comment:**

This work proposes a method for tackling open-world 3D indoor instance segmentation by identifying when objects belong to an "unknown" category and incrementally expanding the set of known classes.  The method uses an auto-labeling scheme to separate known and unknown category labels, and use contrastive clustering to learn prototype representations that will be used for classifying new instances.  The main contribution of this work is the introduction of this problem setup and evaluation protocol (where there are different types of splits for unknown clases) for investigating open-world instance segmentation with unknown classes in 3D.

While this setup is novel and interesting, reviewers have expressed some concerns about the writing and presentation of the work.  Nevertheless, reviewers are overall positive on the work.  The AC agrees that the work has value and can foster additional work in this important direction.

The AC recommends that the authors incorporate clarifications and feedback from the reviewers including:
1. Improvements to the writing to explain the baseline method (PXM7), clarify unclear details in the method (HHFW, L857), design decisions (WNwG), additional visualizations, etc.
2. Add experiments included for the author response (including comparison of techniques for 2D open world)
3. Add a discussion of what are the challenges / insights of this task in 3D vs 2D (PXM7, CNBM, WNwG)
4. Add limitations and societal impact discussion
5. As this is an of growing interest, the AC also recommend the authors include discussion of how this work relates to recent work in open-world 3D scene understanding (many of which are orthogonal to the contributions of this work). A selection of such work is given below:
- Lowis3D: Language-Driven Open-World Instance-Level 3D Scene Understanding (Ding et al. 2023)
- Semantic Abstraction: Open-World 3D Scene Understanding from 2D Vision-Language Models (Ha and Song, CRL 2022)
- CLIP-FO3D: Learning Free Open-world 3D Scene Representations from 2D Dense CLIP (Zhang et al. 2023)
- PointCLIP V2: Adapting CLIP for Powerful 3D Open-world Learning (Zhu et al. 2022)